# ENSO impacts child undernutrition in the global tropics

Jesse K. Anttila-Hughes[1], Amir S. Jina [2,3✉] & Gordon C. McCord [4]

The El Niño Southern Oscillation (ENSO) is a principal component of global climate variability known to influence a host of social and economic outcomes, but its systematic effects on human health remain poorly understood. We estimate ENSO's association with child nutrition at global scale by combining variation in ENSO intensity from 1986-2018 with children's height and weight from 186 surveys conducted in 51 teleconnected countries, containing 48% of the world's under-5 population. Warmer El Niño conditions predict worse child undernutrition in most of the developing world, but better outcomes in the small number of areas where precipitation is positively affected by warmer ENSO. ENSO's contemporaneous effects on child weight loss are detectable years later as decreases in height. This relationship looks similar at both global and regional scale, and has not appreciably weakened over the last four decades. Results imply that almost 6 million additional children were underweight during the 2015 El Niño compared to a counterfactual of neutral ENSO conditions in 2015. This demonstrates a pathway through which human well-being remains subject to predictable climatic processes.

[1] Department of Economics, University of San Francisco, San Francisco, CA, USA. [2] Harris School of Public Policy, University of Chicago, Chicago, IL, USA. [3] National Bureau of Economic Research, Cambridge, MA, USA. [4] School of Global Policy and Strategy, University of California, San Diego, CA, USA. ✉email: amirjina@uchicago.edu

Climate variability is increasingly recognized as a key determinant of health outcomes[1] and a major concern for global climate policy and international public health[2], with the Intergovernmental Panel on Climate Change warning that anthropogenic climate change will very likely increase the frequency and intensity of extreme events[3,4]. The El Niño Southern Oscillation (ENSO) is a major source of climate variability known to affect key social, economic, and health outcomes[5–14]; however, the systematic effects that these correlated shifts in the tropical climate have on global health remain understudied. ENSO's adverse large-scale effects have been documented for hundreds of years[15]. Given that probabilistic forecasts of ENSO have skill at predicting conditions months in advance, there is an opportunity to decouple food insecurity and human nutrition from this predictable climate process. However, analyses of ENSO's impacts on food security have generally focused on a single country or El Niño episode[16,17] and lack global or regional scope to guide national and international public investments that preempt adverse effects of ENSO.

ENSO has destabilizing effects on agriculture[6,15], economic production[7], and social stability[8] throughout areas of the global tropics that are teleconnected to it. It has been linked to human health outcomes directly through its effects on vector- and water-borne infectious diseases[9–13], as well as indirectly by decreasing agricultural yields and increasing food insecurity[14] and the likelihood of conflict[8]. ENSO's adverse effects on yields are particularly acute in the tropics[6], where the vulnerable population of food-insecure children is larger and temperatures are closer to critical crop collapse thresholds[18,19]. Our interest is in the total influence of ENSO variability through all plausible mechanisms—from agricultural productivity to infectious disease to conflict—that are known to affect human nutrition, as well as the systematic differences in ENSO response across places with different precipitation responses to ENSO, across continents and across decades.

This paper estimates ENSO's impacts on human nutrition throughout the global tropics. We leverage over one million child anthropometric records spanning four decades and all developing country regions, building on a growing literature using the Demographic and Health Surveys to document the effects of weather variation on child nutrition[20–23]. We estimate the systematic effect of ENSO-driven tropical climate variability by examining the association between annual eastern equatorial Pacific ENSO state and measures of children's weight from surveys conducted during that year. Our research design estimates the change in nutritional status associated with being in a positive or negative ENSO state compared to a counterfactual of ENSO-neutral conditions. Our results describe shocks to nutritional status rather than identifying the average level of health in a location, which is a complex function of local conditions, such as infrastructure, policies, and the environment. Children's anthropometric measures are sensitive to these nutritional shocks due to their high caloric needs while growing[24,25] and provide a summary measure of contemporary household food security[26]. We find that warmer, more El Niño-like ENSO conditions increase short-term undernutrition in children across the tropics, with the opposite occurring in places where precipitation tends to increase during El Niño. These effects are robust to a wide variety of alternative specifications and approaches and translate into stunting years later.

## Results
### Estimating the global child nutrition effects of ENSO. We capture ENSO variation (Fig. 1a) with the widely used NINO3.4 index of equatorial Pacific sea surface temperature[27–29], which spans 5°N–5°S, 170°W–120°W. Children's weight-for-age z-scores (WAZ) at the time of survey (Fig. 1b) are calculated using the National Center for Health Statistics/Centers for Disease Control and Prevention/World Health Organization (NCHS/CDC/WHO) International Reference Standard[30] intended to provide a single measure of child nutritional outcomes comparable across ages and sexes. We first identify all countries with local climates teleconnected to ENSO (Fig. 1c) for which DHS anthropometric data exist. This yields a sample of 1.3 million children aged 0–4 years interviewed in 186 household surveys between 1986 and 2018. The sample includes 51 countries containing 38% of the world's population and 48% of the world's under-5 population as of 2018. We assign treatment (i.e., the ENSO state when the child was surveyed) annually by tropical year, accounting for typical annual timing in ENSO state change[8,29], by calculating the mean NINO3.4 Sea Surface Temperature (SST) value between May and December of a given year. We assign that to all children interviewed by DHS during that period, as well as all children interviewed during the following year's January–April months, i.e., before the following year's "spring barrier" (see "Methods").

While a warmer ENSO leads to higher temperature throughout the tropics, shifts in precipitation patterns lead to some areas getting wetter than normal while others get drier. We account for potential differences in the effects of ENSO by estimating separate responses in subnational regions where precipitation is positively correlated to warmer ENSO (Fig. 1d) vs. negatively correlated. Since only 6.4% of our sample lives in regions where warmer ENSO leads to clear wet anomalies, we largely focus our discussion on results for the majority of the sample.

The empirical distribution of WAZ is significantly and substantially different ($p < 0.001$) between El Niño and La Niña years, even in the absence of controls (Fig. 2a). A key aspect of our research design, however, rests on exploiting the temporal variability of the ENSO cycle. While ENSO follows a variety of non-random patterns, such as the general progression from El-Niño state to La Niña state, the timing of event occurrence is sufficiently stochastic that even state of the art models have limited prediction skill beyond 6 months into the future[31]. We thus use variation in ENSO anomalies—measured as a deviation from long-run average conditions—in order to statistically isolate the effect of variation in ENSO state on child malnutrition. Following standard practice in the climate impacts literature[32], we purge the estimates of potentially confounding average differences[33,34] between countries and within them based on rural and urban areas using fixed effects/indicator variables for spatial location, detrend the data by major world regions using an annual trend, remove monthly seasonality by major world regions using month fixed effects, and include country-specific controls for mother's age at child's birth and total years of mother's education. Our results correspond to comparing children surveyed at different times in the same country but under different ENSO conditions, while appropriately detrending the data and controlling for average health differences across countries and regions.

### ENSO's effects on contemporaneous nutrition. We estimate that a 1 °C increase in the ENSO index is associated with $0.03\sigma$ ($p = 0.02$) average decrease in WAZ after detrending the data and controlling for location-specific unobservable confounders and mother characteristics (Table 1). We allow the relationship between ENSO and WAZ to vary flexibly (Fig. 2b) and find that the negative association remains across the distribution of ENSO values. The result is substantively similar across a broad range of model specifications (Supplementary Table 1) and across other

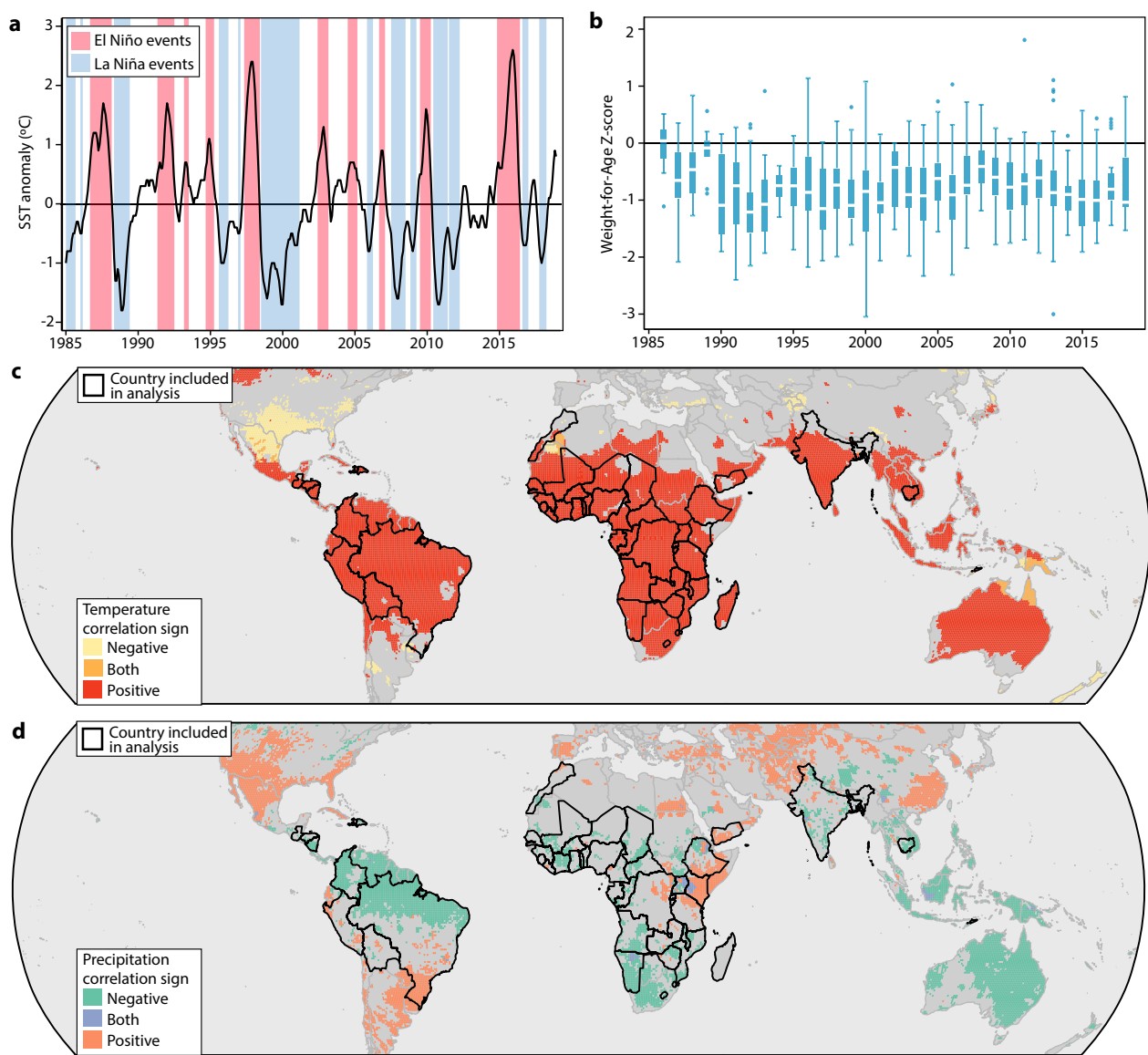

**Fig. 1 Defining the sample of teleconnected children. a** ENSO time series. El Niño or La Niña highlighted in red or blue, respectively. El Niño and La Niña states are defined as follows: when the maximum of a 3-month rolling mean of monthly Oceanic Niño Index (ONI) values is >0.5 °C (Niño-like) or < −0.5 °C (Niña-like) compared to a moving reference climatology following NOAA CPC guidelines[38]. **b** Weight-for-age z-score distribution over time in teleconnected countries (n = 1,253,176 children from 51 surveys). Box plots indicate median (middle white line), 25th, 75th percentile (box), and 5th and 95th percentile (whiskers) as well as outliers (single points). Country composition within each year is different, as a rotating sample of countries is surveyed in each year under the DHS program. **c** Pixel-level monthly correlation of surface temperature (1980–2010) from the UDEL climate dataset and 2-month lag of NINO3.4 Sea Surface Temperature (SST) in teleconnected locations. Teleconnections are defined as pixels where the local temperature shows ≥3 statistically significant months of correlation with the second month lag of NINO3.4 SSTs. Country boundaries indicate sample countries (those having at least 50% of the population living in locations where local temperature is significantly correlated with the second month lag (t − 2) of the NINO3.4 SST index for at least 3 months of the year and with at least two Demographic and Health Surveys measuring anthropometrics). **d** Pixel-level monthly correlation of precipitation (1980–2010) and 2-month lag NINO3.4 SST. There is substantial heterogeneity in how precipitation is affected by ENSO, with areas of both positive and negative correlation. Country boundaries again show sample countries.

outcomes reflecting recent nutrition, including weight-for-height and body mass index (BMI; −0.04σ/°C and p < 0.01 for both measures). Using WHO z-score classification thresholds, warmer ENSO increases the prevalence of underweight (below −2σ in weight-for-age) significantly by 0.6 percentage points per 1 °C (p < 0.05). We find that the risk of wasting (below −2σ in weight-for-height) is similarly positive but not significant (0.3 p.p./°C, p = 0.21), consistent with higher measurement error in height measurements due to the difficulty of measuring child height/length compared to weight[35,36], which decreases the precision of our estimates. All of these patterns are reversed in the minority of

places in our sample (6.4%) where warmer ENSO is correlated to wet anomalies. The heterogeneity in results across regions of wet and dry anomalies points toward the importance of agriculture in mediating the ENSO–nutrition link, though others (e.g., conflict) cannot be ruled out.

**Comparing the 2015 El Niño to large-scale nutrition interventions.** The several degree variation in ENSO cycle implies that it is a meaningful source of variation in population nutrition in the tropics. To give context to the size of these effects, we provide

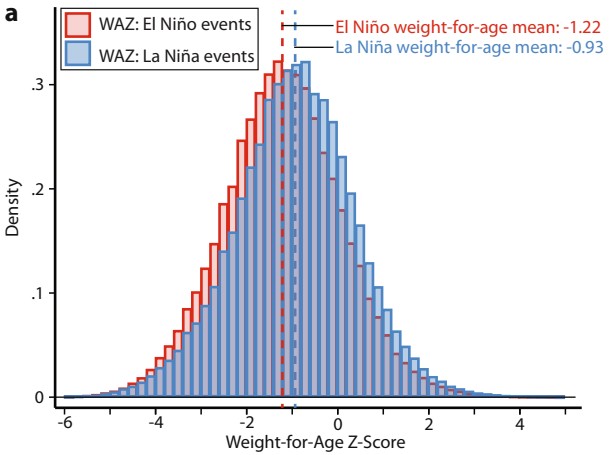

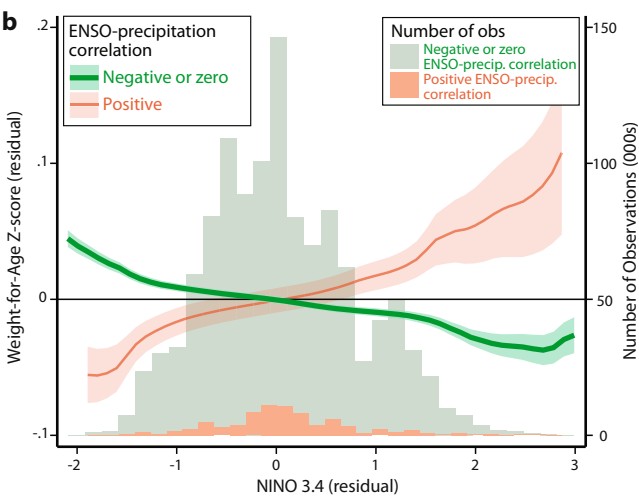

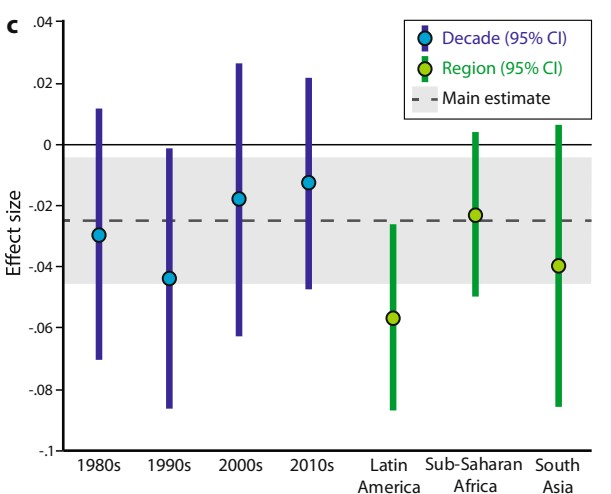

**Fig. 2 Negative effects of ENSO on child weight-for-age. a** Distribution of detrended weight-for-age z-scores (WAZ) during years classified as El Niño (red) and La Niña (blue) according to NOAA definition using NINO3.4 SSTs, with means of each distribution shown. **b** Epanechnikov kernel-weighted local polynomial (bandwidth 0.7) estimate of Table 1. Model 1 showing conditional association of WAZ with ENSO, differentiating countries where precipitation is negatively correlated with ENSO for >3 months in the year over >50% of country area (green) and where it is positively correlated (orange). 95% confidence intervals are shown for the estimated curves. Controls include fixed effects (indicators) for each country; country-specific mother's age at child's birth, total years of mother's education, and rural vs. urban indicator; as well as UNICEF world region-specific linear trends in survey year and fixed effects for the month of interview. Histograms represent the number of observations in each precipitation correlation subsample. **c** Effects of ENSO on WAZ within each decade (blue) and UNICEF world region (green) in the sample (n = 1,253,176 children from 51 surveys), estimated using only locations with non-positive precipitation teleconnections. Dots signify point estimates, bars signify 95% confidence intervals, and gray shaded region and dashed line show main effect from Table 1. The 95% confidence intervals for each decade are: 1980s: [−0.070, 0.011], 1990s: [−0.086, −0.002], 2000s: [−0.063, 0.026], 2010s: [−0.047, 0.021]; and for each region are: Latin America [−0.087, −0.026], Sub-Saharan Africa [−0.050, 0.003], South Asia [−0.086, 0.006].

The human scale of this impact is large given that the under-5 population in our sample countries was 311 million in 2015. By calculating the effect size of the 2015 El Niño summed over all children and dividing by the mean effect size for each nutrition intervention, Fig. 3 shows that offsetting the effects of the 2015 El Niño would require approximately 134 million children receiving multiple micronutrient supplementation (confidence interval (CI) 75–193 million) or 72 million (CI 33–105 million) receiving provision of complementary foods or 72 million (CI 26–118) receiving nutrition education. The effect of the 2015 El Niño is also equivalent to the WAZ reduction from moving 46 million children from urban to rural areas, based on our model results. Using the same 1.92 °C increase and the coefficient in Supplementary Table 5 column 4, the 2015 El Niño increased risk of being below the WHO threshold for underweight by 1.9 percentage points, i.e., an increase of nearly a tenth of the current population rate of 24%. This corresponds to an additional 5.9 million children being driven into underweight status.

**Robustness and implications of ENSO impacts.** Our main result is consistent across alternative specifications, observation weighting, and ENSO variable definitions, as well as across age categories within the sample (Supplementary Tables 1–4). Results are robust to a variety of model specifications controlling for plausible observable and unobservable factors (Supplementary Table 1). While our main results weigh observations so that interpretations are the effect of ENSO on a child in the average country, we also calculate effect on the average child in all sample countries in order to estimate global effects (Fig. 3), and we show that results are consistent if no observation weights are used (Supplementary Table 2). Alternative indicators for ENSO state yield similar results (Supplementary Table 3), with positive deviations from the mean ENSO state decreasing anthropometric z-scores and negative deviations from the mean ENSO state (La Niña events) reducing undernutrition (opposite patterns occur in the few places where precipitation increases with ENSO SST). Coefficients for alternate definitions of ENSO are not statistically distinguishable from our main effect. Supplementary Table 4 estimates the effects of ENSO allowing for different effects by child age categories of 0–5, 6–11, 12–23, 24–35, and

illustrative order-of-magnitude calculations of the scale of public health interventions needed to offset undernutrition on the scale we estimate was caused by the 2015 El Niño, using published effect sizes of nutritional interventions[37]. According to our results, the 1.92 °C increase in the detrended mean NINO3.4 index during the 2015 El Niño event[38], one of the largest on record, likely caused average WAZ in the representative child of our sample countries to decrease by 0.078σ based on the average treatment effect estimated in Supplementary Table 5 column 1.

**Table 1 Anthropometric effects of ENSO.**

| | (1) Weight-for-age | (2) Weight-for-height | (3) Body mass index | (4) Probability below WHO Standard for Underweight | (5) Wasted |
|---|---|---|---|---|---|
| May–December mean NINO 3.4 (°C) | −0.0251** | −0.0377*** | −0.0381*** | 0.00588** | 0.00319 |
| Std. error | (0.0105) | (0.0130) | (0.0132) | (0.00264) | (0.00251) |
| *p* value | 0.0234 | 0.00654 | 0.00674 | 0.0329 | 0.213 |
| 95% CI: lower | −0.0466 | −0.0641 | −0.0649 | 0.00051 | −0.00192 |
| 95% CI: upper | −0.00363 | −0.0113 | −0.0113 | 0.0113 | 0.00829 |
| NINO3.4* I(>50% Pos. Precip.) | 0.0733*** | 0.0489 | 0.0370 | −0.0195*** | −0.00717 |
| *p* value | 0.00239 | 0.114 | 0.270 | 0.000373 | 0.105 |
| Dependent variable mean | −0.931 | −0.228 | −0.0895 | 0.204 | 0.100 |
| Observations | 1,253,176 | 1,205,335 | 1,206,659 | 1,253,176 | 1,205,335 |
| *R*-squared | 0.129 | 0.093 | 0.081 | 0.087 | 0.040 |

Child weight-for-age (1), weight-for-height (2), and body mass index z-scores (3), which all measure shorter-run effects of scarce nutrition, are negatively associated with contemporaneous mean NINO3.4 state (°C), except in areas where precipitation is positively correlated with NINO3.4. Estimates are from OLS regressions with controls consisting of: fixed effects (indicators) for each country; country-specific mother's age at child's birth, total years of mother's education, and rural vs. urban indicator; as well as UNICEF world region-specific linear trends in survey year and fixed effects for the month of interview. Standard errors are two-way clustered at the level of tropical year and subnational administrative unit, and observations are reweighted using DHS sample weights and country size weights in order for estimates to be representative for an average country. (4–5) WHO threshold outcomes show that a warmer ENSO state increases the likelihood of being underweight (below −2σ in weight-for-age) but shows a weaker, statistically insignificant effect on wasting (below −2σ in weight-for-height). Asterisks indicate statistical significance at the 1% (***), 5% (**), and 10% (*) levels (two-sided t test, single hypothesis).

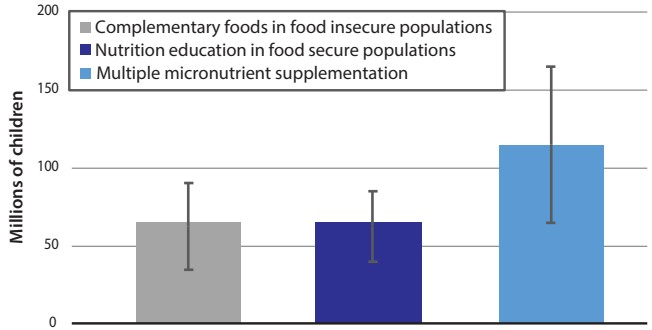

**Fig. 3 Interventions required to offset 2015 El Niño.** Millions of children who must be targeted with specific interventions in order to reverse the effects on malnourishment caused by the 2015 El Niño. Effect sizes calculated using treatment effects in Bhutta et al.[37]. Bars represent the central estimate, with whiskers representing the 95% confidence interval of these estimates. See supplementary information for details of calculation.

36–59 months. With few exceptions, coefficients are consistent in sign and magnitude across age groups for each outcome variable and with the corresponding coefficient in Table 1.

For the purposes of population-wide attribution statistics, we also calculate the average effect of warmer ENSO in the average country, without separating the sample by whether warm ENSO leads to dry or wet anomalies (Supplementary Table 5). The average effect across the sample suggests that warmer ENSO leads to a 0.04σ/°C reduction in weight-for-age ($p = 0.02$), and a 1 percentage point increase in prevalence of underweight ($p < 0.01$). We also test the use of alternative detrending of the data with decade fixed effects to guard against the possibility that results are an artifact of detrending specification (Supplementary Table 6). The lagged effects of ENSO (Supplementary Table 7) indicates no persistent effect of ENSO on child weight-for-age, weight-for-height, or BMI, except in the subsample with positively correlated rainfall. This is consistent with child weight recovering quickly once nutrition returns to adequate levels. On the other hand, child stunting remains affected years after negative shocks from ENSO (Supplementary Table 8), consistent with height being slower to respond to health shocks than weight[20] and with the first 2 years of life being the riskiest period for growth faltering due to scarring[25,39].

Supplementary Tables 9 and 10 show that results are robust to alternative definitions of teleconnection. Supplementary Table 9 extends the main sample to include countries that are teleconnected with NINO3.4 in terms of precipitation but not temperature, while Supplementary Table 10 restricts the sample to include only those countries that have both a significant teleconnection with ENSO via temperatures and precipitation. Supplementary Figs. 2 and 3 and Supplementary Tables 12 and 13 explore whether the ENSO state might affect the timing of DHS surveys within the year and therefore spuriously lead to changes in child anthropometrics due to seasonality. The timing of DHS surveys does not vary as a result of ENSO state (Supplementary Fig. 2 and Supplementary Table 12). ENSO's effect is evident regardless of what time of year the child was surveyed (Supplementary Fig. 3), indicating that effects are not limited to a specific part of the growing or post-harvest season.

Supplementary Table 14 varies the standard error adjustments for serial and spatial autocorrelation and shows that results remain unchanged. Supplementary Table 15 shows that employing logistic regressions for the dichotomous outcome variables results in odd ratios with the same qualitative interpretation as the corresponding linear probability models in Supplementary Table 1. Finally, teleconnected areas remain unchanged across different NINO SST indices (Supplementary Fig. 1) and results hold under a placebo randomization test (Supplementary Fig. 4).

While our estimates of ENSO's effect on child undernutrition are robust, there are nonetheless limitations imposed by both the nature of the data and structure of this research design. DHS data only selectively report migration, making it difficult to deal with any possible migration into or out of the sample that might occur in response to the ENSO cycle. Sufficiently severe ENSO events may also differentially influence the likelihood of being in sample, both at local scale, where, e.g., worse-impacted children may be less likely to end up surveyed due to mortality or illness, and at larger scales, where events such as civil conflict that are known to respond to ENSO[8] may plausibly inhibit the DHS's ability to gather data or ensure data quality. While these aspects of sample selection may lead to unavoidable biases in our results, missing more vulnerable populations would likely bias us away from finding an effect of ENSO on health. Moreover, the consistency of the result across specifications and subsamples suggests that the influence of these limitations on the overall result is likely small.

## Discussion

This analysis measures the total effect of ENSO on child nutrition through all potential measurements and for all affected countries with available data. The negative relationship between child nutrition and warm ENSO state does not appear to vary appreciably across space, with the effects for major world regions in the sample being statistically indistinguishable from our main effect (Fig. 2c). The fact that the effect of a warmer ENSO state varies by the direction of precipitation correlation highlights the importance of precipitation as a mediator in the ENSO–malnutrition relationship. The importance of precipitation may indicate that agriculture plays a strong role in linking ENSO to nutrition outcomes, though we cannot reject the possibility that other channels play important roles in some locations. Other potential channels, such as vector-borne diseases, would likely harm health when precipitation is higher[40], as would impacts of flooding. On the other hand, decreases in rainfall can lead to increased diarrheal disease[41]. Since other channels are not translating into adverse health outcomes for children in places where precipitation increases during the El Niño state, the improved nutrition observed in the data in these locations suggests that increases in agricultural production due to higher rainfall may be a key channel in the global ENSO–nutrition relationship.

We cannot reject that the relationship between child nutrition and ENSO has remained appreciably the same across decades. That there has been little progress in attenuating the food security effects of ENSO, despite increasing incomes and trade connectivity, implies that the limits to adaptation may be strict, at least over the income range of countries in the sample. The international community has set the target of eliminating all forms of malnutrition worldwide by 2030 as part of the Sustainable Development Goals (SDG) agenda and is making efforts to establish metrics, monitor, and implement policies to achieve this goal[42]. Developing countries deemed to be making insufficient progress are being pressured to do more[43]. During 2015–2018, 34% of the children in our sample countries were underweight, implying that in order to meet the hunger SDG the percentage of underweight children would have to decrease by 2.6 percentage points per year. Our estimates suggest that ENSO conditions similar to the 2015 El Niño would eliminate 1 year of progress toward that goal. Indeed, the 2015 El Niño likely played a major role in worsening global hunger, and these results support a view of ENSO driving episodic food insecurity in the tropics where yields are suppressed during El Niño years even while global average yields increase due to gains in the extra-tropics[6].

Hunger continues to affect hundreds of millions of people, with shocks like COVID-19 capable of generating extensive food insecurity and intense suffering. Our work identifies a component of global undernutrition that society can predict, and could contribute to development of hunger early warning systems that would allow actors to deploy nutrition and humanitarian support operations in proactive instead of reactive ways. Governments and agencies engaged in multi-year humanitarian planning and budgetary frameworks should incorporate ENSO forecasts to anticipate fluctuations in availability of local vs. global resource streams needed to ensure progress in fighting malnutrition. Despite scientific progress characterizing ENSO and documenting its various channels of influence on society, much remains to be done to decouple nutrition outcomes in the global tropics from this predictable interannual phenomenon.

## Methods

**Climate data**. ENSO Time Series Data capturing original monthly values of the NINO3.4 index are from the NOAA Center for Weather and Climate Prediction[38]. Local temperature and precipitation data for the teleconnection analysis are obtained from the University of Delaware's terrestrial air temperature and precipitation datasets[44]. Data are provided for monthly temperature and precipitation from 1900 to 2018 on a $0.5° \times 0.5°$ resolution grid of pixels and we use 1950–2014 in our teleconnection analysis.

**Household survey data**. DHS Children's Anthropometric Data are from all DHS surveys containing children's anthropometric data (186 surveys, 51 countries, 1986–2018). We standardize DHS administrative region names, aggregating to supersets if any regions changed borders or split during our sample period.

**Identifying regions teleconnected to ENSO**. We first use the NOAA ENSO time series data to identify regions of the world where temperature or precipitation is teleconnected with ENSO, following prior literature[8]. We designate a country as teleconnected if its temperature is closely coupled to ENSO, defined as having at least 50% of the population living in locations where local temperature as reported in the UDEL global gridded temperature dataset[42] at month $t$ is significantly correlated with the second month lag $(t − 2)$ of the ENSO state (NINO3.4 SST index) for at least 3 months of the year. We designate teleconnection at country level since price effects from ENSO in one subnational region would affect all parts of the country through domestic markets. We note that other NINO indices were tested for this analysis, and teleconnected land areas are largely unchanged, as shown in Supplementary Fig. 1. These teleconnected areas are shown in Fig. 1c.

We perform a similar analysis with precipitation, defining a UDEL pixel as teleconnected if precipitation in month $t$ is significantly correlated with the second month lag $(t − 2)$ of the ENSO state (NINO3.4 SST index) for at least 3 months of the year. These teleconnected areas are shown in Fig. 1d. This allows us to differentiate between places where warmer ENSO leads to dry or wet precipitation anomalies. We separately estimate effects of ENSO in DHS clusters located in first-level administrative units (e.g., state/province) where >50% of land area has ≥3 months per year with a statistically significant positive correlation between precipitation and the second monthly lag of NINO3.4.

Some observations about the patterns of teleconnections are useful. First, we use temperature to define the teleconnected countries (though we alter these assumptions in Supplementary Tables 9 and 10) as the pattern of teleconnections is more contiguous and includes almost all of the areas that also exhibit teleconnections through precipitation. Second, very few places in the world that are teleconnected via temperatures have a decrease in average temperature as the NINO3.4 SST increases. Third, precipitation teleconnections are heterogeneous across space, and we use this heterogeneity in our main analysis, though only 6% of our sample falls into an area that has an increase in rainfall as NINO SSTs increase.

**Selecting a sample of households exposed to ENSO variability**. Through the teleconnection analysis, we choose our main estimating sample. Our treatment variable of interest is the mean value of the NINO3.4 index between May and December in a given year. ENSO events typically begin in late northern hemisphere spring, peak at the year's end, and decay during the early spring of the following year. For this reason, the calendar year is not aligned with exposure to this phenomenon, and specifying exposure that occurred in the early spring of one year would lead to mis-assignment of treatment and biased estimates. We therefore match survey timing from May of year $t$ to April of year $t + 1$ to an ENSO exposure that captures the mean El Niño or La Niña magnitude in a year (occurring between May and December). This ensures that the specific timing of ENSO events is accounted for in our analysis.

Microdata on children's health are from the DHS. We identify all standard DHS surveys from teleconnected countries for which children's anthropometric data are available, generating a sample of 1,253,176 child-level observations in 186 surveys from 51 countries between 1986 and 2018 (Supplementary Table 11). We calculate each child's anthropometrics using their height and weight following the NCHS/CDC/WHO International Reference Standard[26] intended to provide a single measure of child nutritional outcomes comparable across ages and genders and exclude from the analysis the small number of observations (0.56% for WAZ) that are flagged for having improbable values and those with missing values (0.57% for WAZ).

**Econometric estimates of the impact of ENSO on child nutrition**. We estimate the effect of ENSO on anthropometric measure $Y_{ict}$ for child $i$ living in country $c$ in year $t$ (Table 1) using an equation of the form

$$Y_{ict} = \alpha + \beta_n \text{NINO}_t + \beta_p \left[\text{NINO}_t * I\left(\text{Pos}_{\text{Precip}_i}\right)\right] + \gamma \mathbf{X}_{ic} + f(t_{\text{UNICEF}}) + \text{FE}_{cr} + \varepsilon_{ict}, \quad (1)$$

for NINO3.4 anomalies defined as above for the tropical year $t$ in which the anthropometrics for child $i$ were measured, including country-specific controls $\mathbf{X}_{ic}$ for mother's education in years, mother's age at time of child birth, and urban/rural indicators. $f(t)$ captures detrending and seasonality adjustments to the data for each of five major UNICEF world regions using both linear year trends and month of survey fixed effects. We normalize the data by country using fixed effects $\text{FE}_{cr}$ separately identified for rural and urban location within each country. This research design corresponds to comparing detrended differences in child outcomes

within the same country, separately for rural and urban areas, under different global ENSO anomalies. This allows for identification of ENSO's effect under minimal assumptions of potential confounding[23,31]. Standard errors are two-way clustered at the level of interview year ($N = 33$) in order to correct for a common global ENSO shock, as well as at the level of subnational first administrative unit ($N = 532$) in order to adjust for serial correlation in anthropometric indicators over time and space.

Most specifications identify $\beta$, the effect of ENSO, separately for children living in teleconnected areas where El Niño conditions tend to produce wet anomalies ($\beta_P$) and those with neutral or dry anomalies ($\beta_n$). Depending on the specification, the regression weight observations either to produce an estimate of $\beta$ that represents the effect of ENSO on the average country in our sample (that is, using the DHS sampling weights for observations, normalized such that all observations across all surveys sum to unity for each country) or an estimate of $\beta$ that represents the effects on the average child in the countries of our sample (combining normalized DHS sampling weights with population weights for each country). In either case, weights adjust for the fact that countries had different numbers of DHS surveys with different sample sizes over the time period.

For Fig. 2b, we utilize the Frisch–Waugh–Lovell theorem and first residualize our outcome, WAZ, and our independent variable, NINO3.4 SST. That is, we run the following regressions separately for locations with negative/neutral precipitation correlation and positive precipitation correlation, weighted as in Eq. 1:

$$Y_{ict} = \alpha + \gamma \mathbf{X}_{ic} + f(t_{UNICEF}) + FE_{cr} + \varepsilon_{ict} \qquad (2)$$

$$NINO_t = \alpha + \gamma \mathbf{X}_{ic} + f(t_{UNICEF}) + FE_{cr} + \varepsilon_{ict} \qquad (3)$$

We plot the relationship between these residuals using an Epanechnikov kernel-weighted local polynomial regression with a bandwidth of 0.7 in the residualized $x$-variable.

**Reporting summary.** Further information on research design is available in the Nature Research Reporting Summary linked to this article.

## Data availability
The raw survey data are subject to a user agreement and are available at available from the Demographics and Health Surveys Program at https://dhsprogram.com. The raw ENSO data are available via NOAA at https://www.cpc.ncep.noaa.gov/data/indices/. The processed ENSO data are available on Zenodo at https://doi.org/10.5281/zenodo.5208080. The University of Delaware gridded weather data are available at http://climate.geog.udel.edu/climate/html_pages/archive.html. All other datasets produced or used in this analysis can be found at https://doi.org/10.5281/zenodo.5208080.

## Code availability
Data were analyzed using Stata 16, QGIS 2.18, and Matlab 2018b. Code[45] to replicate all results is available on Zenodo at https://doi.org/10.5281/zenodo.5208080.

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

## Acknowledgements

We thank P. Bharadwaj, M. Burke, and T. Garg for helpful comments.

## Author contributions

J.K.A.H., A.S.J., and G.C.M. designed the research, collected the data, analyzed the data, and wrote the paper.

## Competing interests

The authors declare no competing interests.
