## [Peer Review File · Nature Communications]

REVIEWER COMMENTS

Reviewer #1 (Remarks to the Author):

Manuscript#: NCOMMS-20-39794

Introduction

Main claims of the paper

The article evaluates a potential relationship between the El Niño-Southern Oscillation (ENSO) phenomenon and malnutrition in children under the age of five using DHS surveys from 50 countries during 1986-2018. Childhood malnutrition is addressed with Z scores (weight for age, weight for height, and height for age); the article also used body mass index and WHO standards (underweight, wasting, and stunting). ENSO phenomenon is addressed using the El Niño 3.4 index, which is focused on the anomalies during May-December.

The article shows an inverse relationship between the exposure variable known as “the maximum monthly anomaly value of the index during the May-Dec period of a given calendar year” (as an indicator of ENSO occurrence) and the dependent variable related to the children nutrition assessment (Z-scores and WHO standards). The analysis also found that geographically specific areas, where ENSO is positively correlated with more rain, showed a positive relationship between ENSO and children's nutrition Z-scores and WHO standards.

The estimated relationships were robust to different multivariate analysis adjustments, like switching the fixed effects related to countries and month of the DHS survey application. The paper also explored the influence of re-weighting options to balance countries' population, the use of different calendar years for El Niño 3.4 anomalies, the effect on geographically connected countries, and the use of varying clustering strategies to deal with spatial autocorrelation. Generally, all these robustness analyses confirm the main results.

Personal opinion

The paper revealed new evidence about the potential relationship between ENSO and health outcomes. Its methodology incorporated an examination of multiple countries across the globe in support of the fact that a climate phenomenon (ENSO) is related to childhood undernutrition, which is one of today's most critical health issues. This relationship is significant since the climate event studied (ENSO) is a cyclical and predictable phenomenon. Additionally, it reinforces the need for adequate social and health policies to prevent and alleviate adverse effects for children due to ENSO episodes.

The paper contributes to the analysis of children's health outcomes in the context of the ENSO phenomenon. Previous analyses were frequently limited to one specific country. It is less common to find a global analysis that studies the consequences of environmental phenomena on child development. Therefore, this paper includes an analysis of multiple countries using nearly all the data available for them from DHS surveys.

This research could expand the analyses stratifying children by age groups, using wealth index previously constructed by DHS, and using special adjustments to discard effects on variables such as Height for age (HAZ) and stunting. These complementary analyses would be possible by taking advantage of the amount of information already available. Furthermore, it would be beneficial to clarify some statistical procedures employed to calculate parts of the results, such as applying appropriate regression models for categorical outcomes (WHO standards).

Major observations

1. On page 4 (Article File), the authors mentioned: “According to these estimates, the 2.25°C

increase in the Niño 23.4 index during the 2015 El Niño event, one of the largest on record, likely caused average WAZ in the representative child of our sample to decrease by 0.1 DE". There are two questions related to that sentence. First, I am not sure if the grades 2.25* is correct, and it would actually be 2.95* because 2.95* was the max value of Max-Dec of Nino 3.4 index in 2015, so it would be helpful to clarify if 2.25* is the correct increase (see link below). Second, how the authors calculated the potential effect of 0.1 DE? or from which source of information or table could it be inferred?

Link: <https://www.cpc.ncep.noaa.gov/data/indices/sstoi.indices>

2. On page 4 (Article File), the authors mentioned: "Figure 3 shows that around 134 million children would have to receive either multiple micronutrient supplementation (CI 75-193 million), or 72 million (CI 33-105 million) would have to receive provision of complementary foods, or 72 million (CI 26-118) would have to receive nutrition education. The effect of the 2015 El Niño is also equivalent to the WAZ reduction from moving 46 million children from urban to rural areas, based on our model results".

For that section, it is understood that the authors used effect sizes of interventions, previously calculated by Bhutta et al. (2013). However, it would be necessary to know how the authors calculated the specific ranges described and how they did the last estimation related to 46 million children.

3. On page 5 (Article File), the authors mentioned: "The 2015 El Niño increased risk of being below the WHO threshold for undernourished by 2.3 percentage points, i.e., an increase of nearly a tenth of the current population rate of 24%. This corresponds to an additional 7.1 million children being driven into undernourished status". It would be essential to know the specific coefficient (tables) and Max temperature used to estimate the 2.3 percentage points.

4. In table S1, it would be necessary to know which specific adjustment (using the variables listed at the bottom of the table) replicates the main results of table 1. A lector would expect to see the same results of table 1, replicated in column number (4) of table S1; however, they are different, and it was not easy to find the reason that made these results different. I would appreciate it if the authors could clarify this question.

5. It is not clear if the authors had an explicit criterion to include/exclude specific DHS from some years and countries in the analysis. For example, Peru has DHS data for 2004, 2007, 2011, and 2012 to 2019; however, the table S9 showed that Peruvian information included is only from 1992, 1996, and 2000; while for other countries, information is more recent, until 2017 (Senegal) or 2018 (Benin).

6. For all the regression analyses with the WHO standards as the dependent variable, it would be useful to use an appropriate regression method to work with categorical dependent variables. For example, the authors would use a generalized linear model, log-link, and family Poisson or Binomial log-link. Both methods led us to estimate adjusted Prevalence Ratios (aPR). The use of aPR would be opportune in the article. Of course, the authors may prefer other methods that deal with dichotomic outcomes, such as Logit/Probit. However, in this case, it would be better to report the marginal effects (not the immediate coefficients).

Minor observations

7. It would be relevant to know the exact values of confidence intervals for regions shown in Figure 2C.

8. The abstract mentioned 186 surveys, but table S9 (supplementary information) showed a total of 173 surveys.

9. It would be convenient to clarify what specific variables of adjustment related to child characteristics were included. The paper only mentioned "child control variables".

10. In the main text, table 1, and supplementary tables, the WHO standard category for the low weight for age is called "undernourished". However, it would be clearer to call it "underweight" since this term is more usual in the literature. The links below could help.

World Health Organization (WHO) classification of nutritional status of infants and children
<https://www.ncbi.nlm.nih.gov/books/NBK487900/table/fm.s1.t1/>

Nutrition Landscape Information System (NLIS) COUNTRY PROFILE INDICATORS, Interpretation Guide https://www.who.int/nutrition/nlis_interpretation_guide.pdf

Suggestions

The conclusions are coherent with the analysis. However, some additional analyses would reinforce the central hypothesis and enhance the understanding of the relationships studied. In this sense, it would be useful to have the following analyses:

11. Considering we have a large number of observations, around 1,253,173 children with anthropometric measures and covariates without missing values, it would be useful to do a stratified regression by children's age groups. One possible division would be 1) 0-5 months, 2) 6-11 months, 3) 12-23 months, 4) 24-35 months, and 5) 36-59 months. This analysis may offer valuable information about specific groups where ENSO related disasters could have a higher detrimental effect. According to previous studies*, it would be possible to have this kind of differentiated effect by children age groups.

*References:

Del Ninno C, Lundberg M. Treading water: The long-term impact of the 1998 flood on nutrition in Bangladesh. *Economics & Human Biology*. 2005 Mar 1;3(1):67-96.
<https://pubmed.ncbi.nlm.nih.gov/15722263/>

Elorreaga, O.A., Huicho, L. and Lescano, A.G., 2020. El Niño/Southern Oscillation (ENSO) and stunting in children under 5 years in Peru: a double-difference analysis. *The Lancet Global Health*, 8, p.S29.
[https://www.thelancet.com/journals/langlo/article/PIIS2214-109X\(20\)30170-4/fulltext](https://www.thelancet.com/journals/langlo/article/PIIS2214-109X(20)30170-4/fulltext)

Rodriguez-Llanes JM, Ranjan-Dash S, Mukhopadhyay A, Guha-Sapir D. Flood-Exposure is Associated with Higher Prevalence of Child Undernutrition in Rural Eastern India. *International Journal of Environmental Research and Public Health*. 2016 Feb 6;13(2):210.
<https://www.ncbi.nlm.nih.gov/pmc/articles/PMC4772230/>

Rodriguez-Llanes JM, Ranjan-Dash S, Degomme O, Mukhopadhyay A, Guha-Sapir D. Child malnutrition and recurrent flooding in rural eastern India: a community-based survey. *BMJ Open*. 2011 Jan 1;1(2):e000109.
<https://bmjopen.bmj.com/content/1/2/e000109>

12. Furthermore, DHS surveys usually construct a Wealth Index indicator for all the families surveyed. This indicator is estimated using the statistical method of Principal Component Analysis. In the data, the authors could find a variable named "wlthind5" that summarizes all the wealth index analysis and gives a categorical variable with five quintiles (lowest quintile to highest quintile). Although this variable is an indicator of relative wealth within each country's people, it also is a proxy of living conditions (floor and wall materials) and access to public services (electricity and potable water, etc.). Maybe, it would be interesting to run the analyses adjusting for this variable. Likewise, the authors could do stratified regressions for each quintile (or interaction terms). This complementary approach would describe how severe was the ENSO impact depending on the relative wealth of families from each country.

List of countries with wealth index construction in DHS.
<https://dhsprogram.com/topics/wealth-index/Wealth-Index-Construction.cfm>
Steps to constructing the new DHS Wealth Index Shea O. Rutstein, PhD.

https://dhsprogram.com/programming/wealth%20index/Steps_to_constructing_the_new_DHS_Wealth_Index.pdf

13. In table 1 (main text), the authors analyzed the outcome Height for age (HAZ) and Stunting (WHO standard). The results had no statistical significance, and the authors concluded that ENSO conditions do not have a contemporaneous effect on stunting.

To complement this analysis, it would suggest using a similar estimation to table S6 (from Supplementary Information) but considering HAZ and Stunting as outcome variables.

After examining the periodicity of surveys by country (see excel file attached) and considering the most critical ENSO events (ENSO 1997-98 and ENSO 2015-2017) occurred in the period analyzed (1986-2018), it would be appreciated if the authors could do an additional analysis using an independent variable defined as "Two-year lag May-Dec Max Niño 3.4 (*C)", with the dependent variables used in table 1 (HAZ and Stunting). For this analysis, the authors could include only children between 24-59 months of age since younger children (approx. under 24 months) could have not directly experienced the ENSO phenomenon's impact that occurred two years before. Furthermore, variables of adjustment from Table 1 would be included.

It would be useful to know if the coefficient of Z-score height for age (HAZ) and Stunting condition change its statistical significance.

14. It is well known that ENSO has been linked to drought and floods depending on the region. In the article, the interaction term "[NINO3.4 * I(Positive_Precip)]" is not widely discussed, and It would be valuable to know if this interaction term (or a modified one) could be able to represent flooding conditions

In this sense, it would be interesting to use a strategy to identify ENSO related flooding or an estimation to identify an independent effect of ENSO related wet anomalies without reference to what happens in areas with neutral/dry ENSO related anomalies. For this analysis, we could use a sub-sample, including only states/provinces where the authors previously identified the positive correlation between precipitation and the second month lag of Niño 3.4. A similar analysis to table S1 could let us identify the local effect in areas where ENSO produces wet anomalies.

Reviewer: Oliver A. Elorreaga

Reviewer #2 (Remarks to the Author):

This is an interesting and impressive manuscript looking at the impacts of ENSO on childhood undernutrition around the world. It is extremely important to understand the impacts of such climate oscillations on various aspects of human health, and I believe the author's research question has merit. However, when studying the impacts of climate on health it is critical to carefully consider, account for, and discuss the potential mechanisms through which changes in climate can influence the health outcome of interest. I believe the authors have done sound analysis but have not adequately addressed these mechanisms or the motivation and implications of their work. For this to be published, I believe the authors must make major revisions to the framing and discussion of the paper's research questions, methods, and results.

I have laid out specific comments below:

Line 15: "Warmer, drier El Niño conditions" is a bit misleading as it doesn't distinguish between the SST effects of ENSO and the local meteorological effects of ENSO. Throughout the paper the authors should make sure they are being careful about differentiating between SST and teleconnection effects of ENSO.

Line 29-35: I'm unfortunately not convinced by the arguments made here to motivate the analysis. The authors state that "much of humanity is still susceptible to its consequences" when the reality is that we don't yet understand most of these consequences. More importantly, the authors argue that most existing studies focus on single countries and that larger scale studies are needed. However, I would argue that local studies are actually more important for informing context-specific effects and interventions. The relationship between ENSO and any outcome is highly dependent on the impact that ENSO has on local meteorology as well as local infrastructure, wealth, and development. Can the authors expand more on (1) What is the benefit of zooming out spatially? (2) Are there really going to be policies or interventions at the global or multi-national scale? If so, what are these and provide some examples. (3) Is it valid to study a climate phenomenon on such a large spatial scale that has such specific local effects?

Can the authors include more information in the introduction about possible mechanisms explaining the connection between ENSO and malnutrition? Why would we expect to see these effects and what have existing studies found related to their study question? Further, and related to the above comment, how are the authors adding to the existing literature and enhancing our understanding of the mechanistic impacts of ENSO on childhood malnutrition?

Line 45: I'm not sure what the authors mean by "documenting systematic differences in ENSO response within sample."

Why did the authors choose to use Niño 3.4 Index as opposed to other indices (Niño 4, SOI, MEI, ONI)? Different regions of the world have different levels of sensitivity to different ENSO indices. It would be worth doing some sensitivity analyses with other indices to see how the results differ.

Line 57: The authors should further explain the spring barrier delay, why it is important to account for, and how their methods address it.

I'm a little concerned about the annual resolution of the malnutrition outcome. Often teleconnections are felt most strongly in specific seasons, and these timings differ in different countries. Is it possible that there are children's weight measurements that were taken BEFORE any effect of ENSO is felt locally? Would it be possible to stratify the outcome by date of measurement to see if there are stronger effects of ENSO on malnutrition at certain times of the year?

Line 62: This sentence needs citations, or the authors should make it clearer that this is an original analysis.

Line 67: Can the authors discuss what the possible confounders are that they are trying to control? This gets back to the mechanism question and what causal pathway they are trying to isolate.

Also, why are the authors removing monthly seasonality by major world regions instead of by country?

Generally, the structure of the models used need to be discussed more in terms of the mechanisms they are trying to estimate.

Line 91: What do the authors mean by “consistent with higher noise in height measurements, especially for very sick children”? Can they elaborate on this?

Line 95: Please expand on this sentence about agriculture. Again, it is important to fully discuss the mechanistic connections and implications of the results.

Line 98: I would recommend removing this analysis from the paper. The impact of Interventions on malnutrition are extremely dependent on local contexts, so I’m not convinced by the authors applying existing effect estimates from one paper to the entire region of study. I don’t feel that this analysis adds useful or realistic information to the paper and the manuscript would be stronger without it.

Discussion: This section needs much more discussion of the possible mechanisms connecting ENSO to malnutrition that would explain the authors’ results. Additionally, the authors should discuss in more depth how the results could be used to inform specific interventions.

Data: Why are the authors using the maximum monthly Niño 3.4 anomaly value from May - Dec? Can they provide a justification for this choice?

Line 168: How did the authors come up with this method for defining teleconnections? It should be motivated and if possible, citations should be provided to support the use of this method. Also, it is unclear as written if the authors did the teleconnection analysis themselves or relied on previously calculated estimates.

Relatedly, why are the authors defining teleconnection based only on temperature? There are countries that have precipitation teleconnections but not temperature teleconnections.

The specific countries included in the analysis are not stated or shown in the main text. They must be indicated very early in the manuscript so the readers are contextually oriented to the analysis.

Figure 1C needs to show the direction of the association as well as the strength of the association — Why have the authors chosen to use the number of months with associations?

It would be useful to see the countries that they included in Figures 1C and 1D.

Figure 1D should include all countries in the study to demonstrate that most countries have negative precipitation teleconnections

Figure 2C: Can the authors specify in the legend which models produced these estimates?

Supplementary Material: The authors include many sensitivity analyses but it is unclear why they are performing each analysis, especially as many are not discussed in the main text. Each analysis should be motivated and linked to the hypothesized mechanisms and causal pathways.

Reviewer #3 (Remarks to the Author):

I was happy to review this manuscript. It is nice to see another paper in the emerging climate-nutrition literature, and the focus on ENSO appears novel relative to other pertinent studies. The analysis is generally well-executed, with many of my questions about methodology already anticipated in the supplementary materials. The attention to spatial variation is particularly helpful. The paper is nicely written and the results are discussed clearly and with helpful visualizations. In the context of these strengths, I have a few comments and questions:

- Some additional justification for why the focus on ENSO – rather than the more proximate determinants of temperature and precipitation – would be helpful. In other words, if ENSO influences nutrition via changes in temperature and precipitation, why not just model those relationships? I suspect the authors have a good answer, but a sentence or two in the text would help.
- How is the analytic sample influenced by migration (e.g., recent arrivals in the DHS clusters, climate-related out-migration)? Some acknowledgement of the potential (unavoidable) biases would be helpful.
- Additional questions on the analytic sample: How many cases are dropped because of implausible or missing anthropometrics? Does this vary by country or region? Do the authors adjust for measurement error in birth timing (see Larsen et al. 2019 in *Demography*)?
- The authors report that height-for-age did not respond to ENSO (p. 4). However, this is to be expected if they applied the same empirical strategy as for WHZ. I'd recommend they model HAZ among older children (e.g., 3-5 years) as a function of early life ENSO (see Alderman and Headey 2018 in *PLOS* for discussion of timing; and a number of recent papers on climate and nutrition have adjusted their samples and exposure measures accordingly).
- The discussion section refers to statistics on "undernourishment", which are quite different from wasting or stunting. In my view, it would be more helpful to stick strictly to a discussion of global wasting/WHZ (the main outcome in the analysis) or be very clear about the differences between what is measured in the analysis and "undernourishment".
- Invoking "famine" in the discussion is problematic in my view. All of the formally-declared famines in the past few decades have occurred in conflict situations, making attribution to ENSO problematic (in my view). Even in earlier centuries these relationships are problematic. Indeed, Mike Davis – who the authors cite – made the explicit point in *Late Victorian Holocausts* that ENSO only causes mass famine when the social and political conditions allow it.
- I'm surprised by the lack of engagement with the existing literature on climate and nutrition. As just one example - while I appreciate Solomon Hsiang's work - it is inexplicable that he is the most-cited author in this paper when his work has not focused on nutrition to my knowledge. Upshot: please engage with this literature, especially recent papers that have also pooled large numbers of DHS samples to look at similar issues (e.g., Cooper et al. 2019 in *PNAS*, Grace et al. 2015 in *GEC*, Thiede et al. 2020 in *GEC*).

Referee 1

We thank the referee for all the positive comments and support for the paper. They have substantially improved our paper. We address each point individually below (your comments in **bold**, ours in plain text, and excerpts from the paper indented).

Introduction

Main claims of the paper

The article evaluates a potential relationship between the El Niño-Southern Oscillation (ENSO) phenomenon and malnutrition in children under the age of five using DHS surveys from 50 countries during 1986-2018. Childhood malnutrition is addressed with Z scores (weight for age, weight for height, and height for age); the article also used body mass index and WHO standards (underweight, wasting, and stunting). ENSO phenomenon is addressed using the El Niño 3.4 index, which is focused on the anomalies during May-December. The article shows an inverse relationship between the exposure variable known as “the maximum monthly anomaly value of the index during the May-Dec period of a given calendar year” (as an indicator of ENSO occurrence) and the dependent variable related to the children nutrition assessment (Z-scores and WHO standards). The analysis also found that geographically specific areas, where ENSO is positively correlated with more rain, showed a positive relationship between ENSO and children's nutrition Z-scores and WHO standards. The estimated relationships were robust to different multivariate analysis adjustments, like switching the fixed effects related to countries and month of the DHS survey application. The paper also explored the influence of re-weighting options to balance countries' population, the use of different calendar years for El Niño 3.4 anomalies, the effect on geographically connected countries, and the use of varying clustering strategies to deal with spatial autocorrelation. Generally, all these robustness analyses confirm the main results.

Personal opinion

The paper revealed new evidence about the potential relationship between ENSO and health outcomes. Its methodology incorporated an examination of multiple countries across the globe in support of the fact that a climate phenomenon (ENSO) is related to childhood undernutrition, which is one of today's most critical health issues. This relationship is significant since the climate event studied (ENSO) is a cyclical and predictable phenomenon. Additionally, it reinforces the need for adequate social and health policies to prevent and alleviate adverse effects for children due to ENSO episodes. The paper contributes to the analysis of children's health outcomes in the context of the ENSO

phenomenon. Previous analyses were frequently limited to one specific country. It is less common to find a global analysis that studies the consequences of environmental phenomena on child development. Therefore, this paper includes an analysis of multiple countries using nearly all the data available for them from DHS surveys. This research could expand the analyses stratifying children by age groups, using wealth index previously constructed by DHS, and using special adjustments to discard effects on variables such as Height for age (HAZ) and stunting. These complementary analyses would be possible by taking advantage of the amount of information already available. Furthermore, it would be beneficial to clarify some statistical procedures employed to calculate parts of the results, such as applying appropriate regression models for categorical outcomes (WHO standards).

Major observations

1. On page 4 (Article File), the authors mentioned: “According to these estimates, the 2.25°C increase in the Niño 3.4 index during the 2015 El Niño event, one of the largest on record, likely caused average WAZ in the representative child of our sample to decrease by 0.1 DE”. There are two questions related to that sentence. First, I am not sure if the grades 2.25* is correct, and it would actually be 2.95* because 2.95* was the max value of Max-Dec of Niño 3.4 index in 2015, so it would be helpful to clarify if 2.25* is the correct increase (see link below). Second, how the authors calculated the potential effect of 0.1 DE? or from which source of information or table could it be inferred?

Link: <https://www.cpc.ncep.noaa.gov/data/indices/sstoi.indices>

Thank you for pointing this out. There are actually two issues here. First, as you point out, the SST anomaly of the peak of the 2015 when calculated with a 1981-2010 climatology is 2.95°C. However, in our analysis we use a different dataset that detrends the NINO3.4 SST data by using a centered climatology instead of a fixed one. Details are at the link below. This is done to remove the large upward trend in SSTs in the region due to climate change. This detrended dataset is also used by NOAA as an input to the Oceanic Niño Index (ONI). Our main reason to choose this particular dataset for our analysis is that these detrended data and the ONI are used in defining El Niño and La Niña episodes, when thresholds of plus or minus 0.5°C are surpassed. This allows us to maintain a consistent interpretation between the well-known public definition of ENSO episodes and the SST variable that we employ in the analysis. In these data, the November 2015 max is 2.67°C, which occurs in the same month as the 2.95°C in the data that does not have the climate change trends removed. Second, we appreciate you pointing out this value as it alerted us to a labelling error in our draft, where the main model was presented as using the “max” ENSO as opposed to the “mean” ENSO.

We have corrected this in the main text as follows (line 120): “According to our results, the 1.92°C increase in the detrended mean NINO3.4 index during the 2015 El Niño event, one of the largest on record, likely caused average WAZ in the representative child of our sample countries to decrease by 0.078σ based on the average treatment effect estimated in Table S5 column 1.” (We have changed the calculation to use the mean value of 1.92°C. In the data that are not detrended, this value of May-Dec mean NINO3.4 would be 2.05°C.) As shown in Table S3, results are unchanged under the definition of ENSO exposure used.

All datasets mentioned above are located here: <https://www.cpc.ncep.noaa.gov/data/indices/>

This page links both to the sstoi.indices file that you had mentioned in your comment [OISST.v2 (1981-2010 base period): <https://www.cpc.ncep.noaa.gov/data/indices/sstoi.indices>] as well as the detrended data that we have discussed above [ERSSTv5 (centered base periods): http://www.cpc.ncep.noaa.gov/products/analysis_monitoring/ensostuff/detrend.nino34.ascii.txt].

2. On page 4 (Article File), the authors mentioned: “Figure 3 shows that around 134 million children would have to receive either multiple micronutrient supplementation (CI 75-193 million), or 72 million (CI 33-105 million) would have to receive provision of complementary foods, or 72 million (CI 26-118) would have to receive nutrition education. The effect of the 2015 El Niño is also equivalent to the WAZ reduction from moving 46 million children from urban to rural areas, based on our model results”.

For that section, it is understood that the authors used effect sizes of interventions, previously calculated by Bhutta et al. (2013). However, it would be necessary to know how the authors calculated the specific ranges described and how they did the last estimation related to 46 million children.

We apologize for not being more clear in explaining these calculations, and have added a more thorough explanation to the paper, specifically with the following clause (line 125): “By calculating the effect size of the 2015 El Niño summed over all children and dividing by the mean effect size for each nutrition intervention, Figure 3 shows that...”.

Moreover, we add the following text to the SI (line 179):

Calculations for Figure 3. The coefficient on the average treatment effect regression (without splitting the sample by the sign of the precipitation correlation) is -0.04 (see Table S5). The effect of the 1.92°C increase in SST during the 2015 El Niño is therefore $1.92 * -0.04 = -0.078$ in z-score units applied to a total of 310,833,752 children ages 0-4 in the sample countries in 2015 (data from the World Bank’s World Development Indicators). Since the standard deviation of the weight-for-age z-scores

in our sample is 1.51, then the effect size becomes $-0.078/1.51$ or -0.052 , which multiplied by the number of children yields an aggregate change of standardized z-score units of $-16,023,557$.

In the case of provision of complementary foods, Bhutta et al. (2013) review 16 trials and quasi-experimental studies and document significant effects on weight-for-age of 0.26 (0.04-0.48) in standardized mean difference. This leads to an estimate of 62 million (16 million / 0.26) children requiring complementary food interventions to offset the adverse effects of the 2015 El Niño. A similar calculation is done to produce the confidence interval (we use the higher value of 0.48 in effect size CI to provide more conservative estimates on the number of children, corresponding to 33 (16/0.48) million children, and provide a CI of 33-90 million children).

In the case of nutrition education, the Bhutta et al. (2013) review suggests significant effects on weight-for-age z-scores of 0.26 (0.12-0.41) in standardized mean differences. This leads to an estimate of 62 million (16 million / 0.26) food insecure children receiving nutrition education to offset the adverse effects of the 2015 El Niño. A similar calculation is done to produce the confidence interval (we use the higher value of 0.41 in effect size CI, corresponding to 39 (16/0.41) million children, and provide a CI of 39-84 million children).

Finally, in the case of multiple micronutrient supplementation, Bhutta et al. (2013) systematically review 18 trials mostly in developing countries and find an effect size of 0.14, suggesting that around 114 million children (16 million / 0.14) would need to receive multiple micronutrient supplementation to offset the 2015 El Niño. A similar calculation is done to produce the confidence intervals. The effect size CI of multiple micronutrient supplementation is 0.03-0.25. We calculate symmetric CIs for the number of children requiring the public health intervention using the high CI on the intervention effect size, which generates a more conservative (smaller) calculation on the effect of El Niño. In this case, the higher CI value of 0.25 corresponds to 64 million children (16.8/0.25), and we apply the same CI range on the higher number of children to arrive at a CI of 64-165 million children.

3. On page 5 (Article File), the authors mentioned: “The 2015 El Niño increased risk of being below the WHO threshold for undernourished by 2.3 percentage points, i.e., an increase of nearly a tenth of the current population rate of 24%. This corresponds to an additional 7.1 million children being driven into undernourished status”. It would be

essential to know the specific coefficient (tables) and Max temperature used to estimate the 2.3 percentage points.

Thank you for pointing out that we had not clearly identified which particular coefficient was used in this calculation. For this calculation we use the average treatment effect (ATE) results in column (4) of Table S5. The reason to prefer the ATE for this calculation is that it incorporates the net effect of decreases in nutrition in areas where NINO3.4 temperatures are negatively correlated with local precipitation and increases in areas where there is a positive correlation. This helps with the ease of interpretation. Based on comment 1, we have also recalculated this to reflect our correction in the average temperature value 2015 El Niño. The 1.92°C increase in NINO3.4 during the 2015 El Niño would increase each child's risk of being underweight by 1.9 (0.0101*1.92) percentage points. This corresponds to an additional 5.9 million (0.019 * 311 million) children being driven into underweight status.

We've amended the text to make the choice of regression coefficients clear (lines 131): "Using the same 1.92°C increase and the coefficient in Table S5 column 4, the 2015 El Niño increased risk of being below the WHO threshold for underweight by 1.9 percentage points, i.e., an increase of nearly a tenth of the current population rate of 24%. This corresponds to an additional 5.9 million children being driven into underweight status."

4. In table S1, it would be necessary to know which specific adjustment (using the variables listed at the bottom of the table) replicates the main results of table 1. A lector would expect to see the same results of table 1, replicated in column number (4) of table S1; however, they are different, and it was not easy to find the reason that made these results different. I would appreciate it if the authors could clarify this question.

We apologize for the lack of clarity. We have now added to Table S1 a new column (1) which replicates the result in Table 1, for comparison to the specifications in the other columns.

	(1)	(2)	(3)	(4)	(5)	(6)	(7)
Panel A: Weight for Age							
May-Dec Mean Niño 3.4 (°C)	-0.0251**	-0.0260**	-0.0294**	-0.0302***	-0.0181*	-0.0284***	-0.0276***
std. error:	(0.0105)	(0.0126)	(0.0116)	(0.0104)	(0.00931)	(0.00984)	(0.00939)
P-value:	0.0234	0.0476	0.0162	0.00637	0.0605	0.00694	0.00600
95% Ci: lower	-0.0466	-0.0516	-0.0529	-0.0513	-0.0371	-0.0484	-0.0468
95% Ci: upper	-0.00363	-0.000294	-0.00580	-0.00914	0.000847	-0.00835	-0.00851
* > 50% Pos. Precip. Corr.)	0.0733***	0.184**	0.0652**	0.0683***	0.0592**	0.0718***	0.107**
std. error:	(0.0222)	(0.0741)	(0.0260)	(0.0239)	(0.0258)	(0.0223)	(0.0408)
P-value:	0.00239	0.0184	0.0176	0.00747	0.0285	0.00298	0.0130
95% Ci: lower	0.0280	0.0331	0.0122	0.0196	0.00665	0.0263	0.0242
95% Ci: upper	0.119	0.335	0.118	0.117	0.112	0.117	0.190
R-squared	0.129	0.099	0.125	0.127	0.132	0.130	0.139
Dependent variable mean	-0.931	-0.941	-0.931	-0.931	-0.931	-0.931	-0.931
Panel B: Underweight (low w4a)							
May-Dec Mean Niño 3.4 (°C)	0.00588**	0.00537	0.00604*	0.00714**	0.00409*	0.00646**	0.00571**
std. error:	(0.00264)	(0.00340)	(0.00301)	(0.00263)	(0.00226)	(0.00239)	(0.00240)
P-value:	0.0329	0.124	0.0536	0.0107	0.0796	0.0110	0.0233
95% Ci: lower	0.000510	-0.00156	-0.000101	0.00178	-0.000511	0.00158	0.000827
95% Ci: upper	0.0113	0.0123	0.0122	0.0125	0.00868	0.0113	0.0106
* > 50% Pos. Precip. Corr.)	-0.0195***	-0.0539**	-0.0188***	-0.0187***	-0.0178***	-0.0191***	-0.0278**
std. error:	(0.00489)	(0.0215)	(0.00637)	(0.00532)	(0.00468)	(0.00502)	(0.0110)
P-value:	0.000373	0.0173	0.00592	0.00136	0.000597	0.000623	0.0169
95% Ci: lower	-0.0294	-0.0976	-0.0318	-0.0295	-0.0274	-0.0293	-0.0503
95% Ci: upper	-0.00949	-0.0101	-0.00581	-0.00783	-0.00830	-0.00883	-0.00533
R-squared	0.087	0.069	0.083	0.085	0.089	0.087	0.094
Dependent variable mean	0.204	0.205	0.204	0.204	0.204	0.204	0.204
Observations	1,253,176	1,268,295	1,253,176	1,253,176	1,253,173	1,253,176	1,253,176
Country FE	yes	yes	yes	yes	yes	yes	yes
Yeartrend	By Region	yes	yes	By Region	By Region	By Region	By Region
Interview month FE	By Region	yes	yes	yes	By Country	yes	yes
Rural FE	By Country		yes	By Country	By Country	By Country	By Country
Child & Mother controls	By Country		yes	yes	yes	By Country	yes
Admin1 FE							yes

5. It is not clear if the authors had an explicit criterion to include/exclude specific DHS from some years and countries in the analysis. For example, Peru has DHS data for 2004, 2007, 2011, and 2012 to 2019; however, the table S9 showed that Peruvian information included is only from 1992, 1996, and 2000; while for other countries, information is more recent, until 2017 (Senegal) or 2018 (Benin).

All of the surveys mentioned by the referee for Peru, Senegal and Benin were included in our analysis. Due to a mistake in the construction of Table S9 in the previous draft, several surveys were not listed despite being part of our analysis. We have corrected the table in the paper (it is now Table S11) and included it below. We thank the referee for noticing the omission.

Country	Survey Years									
East Asia & Pacific										
Cambodia	2000	2005	2010	2014						
Timor Leste	2009	2016								
Latin America & Caribbean										
Bolivia	1989	1994	1998	2003	2008					
Brazil	1986	1996								
Colombia	1986	1995	2000	2005	2010					
Dominican Republic	1986	1991	1996	2002	2007	2013				
Guatemala	1987	1995	1999	2015						
Haiti	1995	2000	2006	2012	2017					
Honduras	2006	2012								
Nicaragua	1998	2001								
Peru	1992	1996	2000	2005	2008	2009	2010	2011	2012	
Middle East & North Africa										
Morocco	1987	1992	2003							
Yemen	1992									
sub-Saharan Africa										
Angola	2016									
Benin	1996	2001	2006	2012	2018					
Burkina Faso	1993	1999	2003	2010						
Burundi	2010									
Cameroon	1991	1998	2004	2011						
Central African Republic	1994									
Chad	1997	2004	2015							
Comoros	1996	2012								
Congo, Rep.	2005	2012								
Congo, Dem. Rep.	2007	2014								
Cote d'Ivoire	1994	1998	2012							
Ethiopia	2000	2005	2010	2016						
Gabon	2000	2012								
Ghana	1988	1993	1998	2003	2008	2014				
Guinea	1999	2005								
Kenya	1993	1998	2003	2008	2014					
Lesotho	2004	2009	2014							
Liberia	2007									
Madagascar	1992	1997	2004							
Malawi	1992	2000	2004	2010	2016					
Mali	1987	1996	2001	2006	2013					
Mozambique	1997	2003	2011							
Namibia	1992	2000	2006	2013						
Niger	1992	1998	2006	2012						
Nigeria	1990	1999	2003	2008	2013					
Rwanda	1992	2000	2005	2010	2015					
São Tomé e Príncipe	2009									
Senegal	1986	1993	2005	2010	2013	2014	2015	2016	2017	
Sierra Leone	2008	2013								
Swaziland	2006									
Tanzania	1992	1996	1999	2005	2010	2016				
Togo	1988	1998	2014							
Uganda	1989	1995	2000	2006	2011	2014				
Zambia	1992	1996	2002	2007	2014					
Zimbabwe	1988	1994	1999	2005	2010	2015				
South Asia										
Bangladesh	1997	2000	2004	2007	2011	2014				
India	1993	1999	2006	2016						
Maldives	2009	2017								

6. For all the regression analyses with the WHO standards as the dependent variable, it would be useful to use an appropriate regression method to work with categorical dependent variables. For example, the authors would use a generalized linear model, log-link, and family Poisson or Binomial log-link. Both methods led us to estimate adjusted Prevalence Ratios (aPR). The use of aPR would be opportune in the article. Of course, the

authors may prefer other methods that deal with dichotomic outcomes, such as Logit/Probit. However, in this case, it would be better to report the marginal effects (not the immediate coefficients).

For our analysis on dichotomous outcomes, we opt for a linear probability model (LPM) because most of our specifications include interaction terms (specifically, interacting with the sign of the precipitation correlation). These interactions in combination with location fixed effects complicate estimation and interpretation using nonlinear models such as proportional hazard or Poisson¹. The LPM is often used in large-data health studies with dichotomous outcomes.^{2,3}

As a robustness check, we follow the referee's request and implement a logistic version of our main regression specification to recover odds ratios, and show in the table below that results are not qualitatively different. Columns 1 and 3 are the results in Table 1 of the paper using the linear probability model for the underweight and wasted binary outcomes. Columns 2 and 4 show results for the corresponding logit specifications, reporting odds ratios. The results are qualitatively unchanged from the LPM results. Given the easier interpretability of the LPM in the presence of interactions with the treatment variable, and the fact that for the conclusion the coefficients allow for easier calculation of population-level impact of a particular El Niño on numbers of children pushed into underweight status, we opt to leave the LPM result in the main paper and have added the table below with the logit results as Table S15. We would of course be happy to move the logit results into the main paper if the referee and editor would prefer.

We describe the new results in the SI text as follows (line 154):

Table S15 implements a logistic version of regressions 4-5 in Table 1 which have dichotomous dependent variables. Columns (1) and (3) are the results in Table 1 of the paper using the linear probability model for the underweight, wasted and stunted binary outcomes. Columns (2) and (4) show results for the corresponding logit specifications, reporting odds ratios. The results are qualitatively unchanged from the LPM results.

¹ Ai C, Norton EC. Interaction terms in logit and probit models. *Economics Letters* 2003; 80[1]:123–129.

² Demombynes G, Trommlerová SK. What has driven the decline of infant mortality in Kenya in the 2000s? *Economics & Human Biology* 2016; 21:17–32.

³ Pongou R. Why is infant mortality higher in boys than in girls? A new hypothesis based on preconception environment and evidence from a large sample of twins. *Demography* 2013; 50[2]:421–444.
<https://doi.org/10.1007/s13524-012-0161-5> PMID: 23151996

	(1)	(2)	(3)	(4)
	Probability below WHO Standard for:			
	underweight		wasted	
	LPM	Logit	LPM	Logit
May-Dec Mean Niño 3.4 (°C)	0.00588**	1.033**	0.00319	1.037*
std. error:	(0.00264)	(0.0148)	(0.00251)	(0.0205)
p-value:	0.0329	0.0224	0.213	0.0671
95% CI: lower	0.00051	1.005	-0.00192	0.997
95% CI: upper	0.0113	1.063	0.00829	1.078
* I(> 50% Pos. Precip. Corr.)	-0.0195***	0.881***	-0.00717	0.908
p-value:	0.000373	0.00725	0.105	0.103
Dependent variable mean:	0.204	0.204	0.100	0.100
Observations	1,253,176	1,253,176	1,205,335	1,203,154
R-squared	0.087		0.040	

Minor observations

7. It would be relevant to know the exact values of confidence intervals for regions shown in Figure 2C.

We apologise for only presenting this as a figure and not also including the confidence intervals in the text. We have amended the caption of the figure with the following text that now details the confidence intervals:

Effect of ENSO on WAZ for each decade and UNICEF world region in the sample, excluding locations with positive precipitation teleconnections. Dots signify point estimates, bars signify 95% confidence intervals, and grey shaded region and dashed line show main effect from Table 1. The 95% confidence intervals for each decade are: 1980s: [-0.070, 0.011], 1990s: [-0.086, -0.002], 2000s: [-0.063, 0.026], 2010s: [-0.047, 0.021]; and for each region are: Latin America [-0.087, -0.026], Sub-Saharan Africa [-0.050, 0.003], South Asia [-0.086, 0.006].

8. The abstract mentioned 186 surveys, but table S9 (supplementary information) showed a total of 173 surveys.

We thank you for catching the inconsistency. Due to an error in constructing Table S9 in the previous draft, the table was missing some of the surveys in our data. We have corrected the table (see below, now Table S11), such that all 186 surveys are now listed.

9. It would be convenient to clarify what specific variables of adjustment related to child characteristics were included. The paper only mentioned “child control variables”.

We apologize for not being sufficiently clear. All controls for regressions are listed in the table captions, and in the case of child- and mother-level controls these include mother’s age at child’s birth and mother’s total years of education attained. We now explicitly say so in the text and captions, specifically in the sentence on the SI that previously said “child control variables” and now reads “Model (3) normalizes data by rural/urban location and adds child- and mother-level controls (mother’s age at child’s birth, total years of mother’s education), resulting in a decrease in the coefficient for positive precipitation correlation locations but no change for other locations.” (line 16)

We note that the dependent variables are age-adjusted z-scores, so implicitly age is adjusted for in these specifications. What becomes relevant, then, is whether there are varying effects of ENSO by age, which is what the referee asks below in question 11.

Finally, we directly test whether our results are robust to including more child-level controls such as age and sex in the table below. The column on the left is the main result in the paper on weight for age in Table 1; column (2) adds country-specific controls for child age and child sex. The result indicates that adding these controls does not change the results.

	(1)	(2)
	Weight for age	Weight for age
May-Dec Mean Niño 3.4 (°C)	-0.0251**	-0.0287**
std. error:	(0.0105)	(0.0113)
p-value:	0.0234	0.0158
95% CI: lower	-0.0466	-0.0517
95% CI: upper	-0.00363	-0.00576
* I(>50% Pos. Precip. Corr.)	0.0733***	0.0767***
p-value:	0.00239	0.00210
Dependent variable mean:	-1.078	-0.371
Observations	1,253,176	1,205,335
R-squared	0.129	0.159

10. In the main text, table 1, and supplementary tables, the WHO standard category for the low weight for age is called “undernourished”. However, it would be clearer to call it “underweight” since this term is more usual in the literature. The links below could help.

World Health Organization (WHO) classification of nutritional status of infants and children

<https://www.ncbi.nlm.nih.gov/books/NBK487900/table/fm.s1.t1/>

Nutrition Landscape Information System (NLIS) COUNTRY PROFILE INDICATORS, Interpretation Guide https://www.who.int/nutrition/nlis_interpretation_guide.pdf

We thank the referee for pointing out this discrepancy in our use of the WHO terminology. We have made our use of anthropometric terms consistent throughout the text, using “underweight” to refer to children whose height for age z-score is below -2.

Suggestions

The conclusions are coherent with the analysis. However, some additional analyses would reinforce the central hypothesis and enhance the understanding of the relationships studied. In this sense, it would be useful to have the following analyses:

11. Considering we have a large number of observations, around 1,253,173 children with anthropometric measures and covariates without missing values, it would be useful to do a stratified regression by children's age groups. One possible division would be 1) 0-5 months, 2) 6-11 months, 3) 12-23 months, 4) 24-35 months, and 5) 36-59 months. This analysis may offer valuable information about specific groups where ENSO related disasters could have a higher detrimental effect. According to previous studies*, it would be possible to have this kind of differentiated effect by children age groups.

We thank the referee for asking us to explore whether effects of ENSO vary by child age. Table S4 in the SI now shows the results of this decomposition, estimating group-level versions of our main interacted results from Table 1 in the main text. We find broadly similar patterns as in our main results, with most children in most age groups showing weight loss responses similar to our main results. Note that estimates have less precision given the smaller sample size within each group. We further explore age-differentiated effects in our response to comment 13 below on ENSO conditions around time of birth and evidence of stunting, and now discuss both results in the main text and SI. In the main text we have added (line 138): “...and across age categories within the sample (Tables S1-S4).” We have also expanded our citations on prior ENSO effects in line with suggestions below. In the SI we have added (line 73):

“Table S4 estimates effects of ENSO on the same anthropometric measures as Table 1, but allowing for different effects by child age categories of 0-5 months, 6-11 months, 12-23 months, 24-35 months and 36-59 months. With few exceptions, coefficients are consistent in sign and magnitude across age groups for each outcome variable and with the corresponding coefficient in Table 1. The significantly smaller sample size in each

age category results in imprecise estimates, however. The 36-59 months category has a larger sample and correspondingly higher precision in the estimates.”

		(1) Weight for age	(2) Weight for height	(3) Body mass index	(4) Probability below WHO Standard for: underweight	(5) Standard for: wasted
Age 0-5 months (column 1 N = 134,618)	May-Dec Mean Niño 3.4 (°C)	0.00285	-0.0305	-0.0385	-0.00268	-0.00235
	P value:	0.978	0.412	0.572	0.851	0.828
	* I(> 50% Pos. Precip. Corr.)	-0.0776	-0.128	-0.282	0.054	0.0378
	P value:	0.609	0.307	0.212	0.136	0.261
Age 6-11 months (column 1 N = 144,904)	May-Dec Mean Niño 3.4 (°C)	-0.0275	-0.0567	-0.0677	0.00238	0.00448
	P value:	0.394	0.322	0.465	0.586	0.718
	* I(> 50% Pos. Precip. Corr.)	-0.00519	-0.0489	-0.108	0.0231	0.0154
	P value:	0.962	0.688	0.541	0.501	0.64
Age 12-23 months (column 1 N = 272,753)	May-Dec Mean Niño 3.4 (°C)	-0.0253	-0.0207	-0.0203	0.00694	-0.00102
	P value:	0.256	0.558	0.342	0.206	0.865
	* I(> 50% Pos. Precip. Corr.)	0.115**	0.0562	0.0963	-0.0184**	-0.0029
	P value:	0.0213	0.309	0.115	0.0321	0.706
Age 24-35 months (column 1 N = 256,897)	May-Dec Mean Niño 3.4 (°C)	-0.0356	-0.00629	0.00147	0.0109	-0.00113
	P value:	0.177	0.862	0.98	0.116	0.851
	* I(> 50% Pos. Precip. Corr.)	0.0759*	0.105	0.139	-0.0201*	-0.0153
	P value:	0.0658	0.169	0.191	0.0872	0.264
Age 36-59 months (column 1 N = 443,489)	May-Dec Mean Niño 3.4 (°C)	-0.0268	-0.0734**	-0.0726**	0.00642	0.0126
	P value:	0.410	0.0371	0.0474	0.205	0.148
	* I(> 50% Pos. Precip. Corr.)	0.118	0.105**	0.0941	-0.0550**	-0.0274**
	P value:	0.113	0.0496	0.111	0.0336	0.0367
Dependent variable mean:		-0.931	-0.228	-0.0895	0.204	0.1
Observations		1,252,661	1,204,826	1,206,146	1,252,661	1,204,826
R-squared		0.129	0.093	0.081	0.087	0.04

***References:**

Del Ninno C, Lundberg M. Treading water: The long-term impact of the 1998 flood on nutrition in Bangladesh. *Economics & Human Biology*. 2005 Mar 1;3(1):67–96.
<https://pubmed.ncbi.nlm.nih.gov/15722263/>

Elorreaga, O.A., Huicho, L. and Lescano, A.G., 2020. El Niño/Southern Oscillation (ENSO) and stunting in children under 5 years in Peru: a double-difference analysis. *The Lancet Global Health*, 8, p.S29.
[https://www.thelancet.com/journals/langlo/article/PIIS2214-109X\(20\)30170-4/fulltext](https://www.thelancet.com/journals/langlo/article/PIIS2214-109X(20)30170-4/fulltext)

Rodriguez-Llanes JM, Ranjan-Dash S, Mukhopadhyay A, Guha-Sapir D. Flood-Exposure is Associated with Higher Prevalence of Child Undernutrition in Rural Eastern India. *International Journal of Environmental Research and Public Health*. 2016 Feb 6;13(2):210.
<https://www.ncbi.nlm.nih.gov/pmc/articles/PMC4772230/>

Rodriguez-Llanes JM, Ranjan-Dash S, Degomme O, Mukhopadhyay A, Guha-Sapir D. Child malnutrition and recurrent flooding in rural eastern India: a community-based survey. *BMJ Open*. 2011 Jan 1;1(2):e000109.

<https://bmjopen.bmj.com/content/1/2/e000109>

12. Furthermore, DHS surveys usually construct a Wealth Index indicator for all the families surveyed. This indicator is estimated using the statistical method of Principal Component Analysis. In the data, the authors could find a variable named "wlthind5" that summarizes all the wealth index analysis and gives a categorical variable with five quintiles (lowest quintile to highest quintile). Although this variable is an indicator of relative wealth within each country's people, it also is a proxy of living conditions (floor and wall materials) and access to public services (electricity and potable water, etc.). Maybe, it would be interesting to run the analyses adjusting for this variable. Likewise, the authors could do stratified regressions for each quintile (or interaction terms). This complementary approach would describe how severe was the ENSO impact depending on the relative wealth of families from each country.

List of countries with wealth index construction in DHS.

<https://dhsprogram.com/topics/wealth-index/Wealth-Index-Construction.cfm>

Steps to constructing the new DHS Wealth Index Shea O. Rutstein, PhD.

https://dhsprogram.com/programming/wealth%20index/Steps_to_constructing_the_new_DHS_Wealth_Index.pdf

We thank the referee for the idea of using the wealth index as a control or to detect varying effects of ENSO across socioeconomic groups. Since the wealth index is constructed as quintiles at survey level, it is not designed for comparison across surveys either in the same country and especially not across countries. In fact, the DHS Methodological report explicitly says that the wealth index should not be used in this way:

While the DHS wealth index is useful for analyzing differentials in economic status within countries, for the purpose of exploring issues of economic equity and poverty, it should be emphasized that the wealth index is constructed as a relative index within each country at the time of the survey. Each wealth index has a mean value of zero and a standard deviation of one. Thus, specific scores cannot be directly compared across countries or over time. For example, in an extremely poor country a household may be included in the highest wealth quintile but is not necessarily well-off in absolute terms.⁴

We note that the wealth index is only available for two-thirds of our sample, which means that using it in the specification significantly reduces power. We also note that our specifications include country-specific controls for whether the household is rural or urban, and country-specific controls for mother's level of education, which is nearly universally reported. Taken

⁴ <https://www.dhsprogram.com/pubs/pdf/MR9/MR9.pdf>

together, these variables proxy for much of the within-country variation in socioeconomic status that the wealth index would pick up.

For completeness, the table below shows the results in the main specifications of Table 1, controlling for the household's wealth index interacted with the specific DHS survey, since the index should not be compared across surveys or countries. Moreover, the wealth index of the household may itself be affected by the ENSO shock, thus making it a problematic control variable. Nevertheless, the results below show that the coefficients are very similar to those in Table 1. Due to these restrictions and since the number of observations is reduced by around 25% thus reducing precision of the estimates, we opt against controlling for the wealth index in our specifications.

	(1)	(2)	(3)	(4)	(5) (6) (7)		
	Weight for age	Weight for height	Body mass index	Height for age	Probability below WHO Standard for:		
					undemourished	wasted	stunted
May-Dec Mean Niño 3.4 (*C)	-0.0371*	-0.0348	-0.0304	-0.0136	0.00910*	0.00632**	0.00278
std. error:	(0.0180)	(0.0209)	(0.0210)	(0.0190)	(0.00513)	(0.00261)	(0.00330)
P value:	0.0574	0.117	0.168	0.486	0.0960	0.0287	0.413
95% CI: lower	-0.0755	-0.0793	-0.0752	-0.0541	-0.00182	0.000754	-0.00426
95% CI: upper	0.00132	0.00977	0.0143	0.0269	0.0200	0.0119	0.00982
* I(> 50% Pos. Precip. Corr.)	0.0103	-0.0845	-0.109	0.107***	-0.00599	0.0111	-0.0278***
P value:	0.822	0.193	0.115	0.00225	0.411	0.183	0.00573
Dependent variable mean:	-0.884	-0.198	-0.0645	-1.276	0.189	0.0971	0.331
Observations	847,536	821,154	821,981	832,763	847,536	821,154	832,763
R-squared	0.134	0.096	0.085	0.076	0.092	0.043	0.083

13a. In table 1 (main text), the authors analyzed the outcome Height for age (HAZ) and Stunting (WHO standard). The results had no statistical significance, and the authors concluded that ENSO conditions do not have a contemporaneous effect on stunting. To complement this analysis, it would suggest using a similar estimation to table S6 (from Supplementary Information) but considering HAZ and Stunting as outcome variables. 13b. After examining the periodicity of surveys by country (see excel file attached) and considering the most critical ENSO events (ENSO 1997-98 and ENSO 2015-2017) occurred in the period analyzed (1986-2018), it would be appreciated if the authors could do an additional analysis using an independent variable defined as "Two-year lag May-Dec Max Niño 3.4 (*C)", with the dependent variables used in table 1 (HAZ and Stunting). For this analysis, the authors could include only children between 24-59 months of age since younger children (approx. under 24 months) could have not directly experienced the ENSO phenomenon's impact that occurred two years before. Furthermore, variables of adjustment from Table 1 would be included. It would be useful to know if the coefficient of Z-score height for age (HAZ) and Stunting condition change its statistical significance.

We greatly appreciate this suggestion (also suggested by referee 3) which speaks to both taking the dynamic aspects of these results as well as the extant literature very seriously. We now include Table S8 in the SI which estimates the HAZ and stunting effects of early life exposure to ENSO in children of 24-59 months of age. Specifically, we assign birth dates to tropical year using the same May-April spring barrier assignment used for interview dates, and estimate a version of our main precipitation teleconnection-differentiated model where HAZ and stunting are a function of ENSO state during the year of birth as well as the subsequent two years of the child's life.

Notably, children experiencing warmer, more El-Niño-like conditions during their year of birth translates into a higher likelihood of stunting and lower height for age Z-scores, which respond to ENSO state not only in the year of birth but also in the two years that follow, albeit at somewhat diminished effect size. This effect is mirrored and opposite in the small number of positively teleconnected regions only during the first year of life. These results substantially strengthen the evidence of ENSO's malnutrition effects and highlight their persistent nature, which we now highlight in the paper, and we are very grateful to the referees for the suggestion to further explore the delayed effects on HAZ and stunting.

		(1) Height for age	(2) Probability below WHO Standard for stunted
Year of birth	May-Dec Mean Niño 3.4 (°C)	-0.0327***	0.00903**
	P-value:	0.00890	0.0351
	* I(> 50% Pos. Precip. Corr.)	0.0569**	-0.00157
	P-value:	0.0349	0.821
Year after birth	May-Dec Mean Niño 3.4 (°C)	-0.0213*	0.00724*
	P-value:	0.0668	0.0838
	* I(> 50% Pos. Precip. Corr.)	-0.0248	0.00913
	P-value:	0.590	0.386
Two years after birth	May-Dec Mean Niño 3.4 (°C)	-0.0244**	0.00774*
	P-value:	0.0266	0.0610
	* I(> 50% Pos. Precip. Corr.)	-0.00408	0.00325
	P-value:	0.906	0.711
Dependent variable mean:		-1.713	0.417
Observations		664,658	664,658
R-squared		0.136	0.120

The changes to the text are as follows. In the abstract we added the following (line 17): “ENSO’s contemporaneous effects on child weight loss are detectable years later as decreases in height.” In the main text, we added the following in the Results (line 161): “This is consistent with child weight recovering quickly once nutrition returns to adequate levels. On the other hand, child stunting remains affected years after negative shocks from ENSO (Table S8), consistent with height being slower to respond to health shocks than weight and with the first

two years of life being the riskiest period for growth faltering due to scarring (Almond and Currie 2011, Alderman and Headley 2018).” Finally, in the SI we describe the new table as follows (line 88): “Table S8 shows evidence for the persistent effects of ENSO by focusing on older children (ages 2-5 years) and the effect of the ENSO state during year of birth and during two subsequent years (Alderman and Headley 2018) on the height for age and likelihood of being stunted. While weight can recover quickly after a negative shock, child height is a slower-moving cumulative anthropometric indicator and shows the longer-lasting effects of ENSO state on child nutrition during early life.”

14. It is well known that ENSO has been linked to drought and floods depending on the region. In the article, the interaction term "[NINO3.4 * I(Positive_Precip)]" is not widely discussed, and It would be valuable to know if this interaction term (or a modified one) could be able to represent flooding conditions

In this sense, it would be interesting to use a strategy to identify ENSO related flooding or an estimation to identify an independent effect of ENSO related wet anomalies without reference to what happens in areas with neutral/dry ENSO related anomalies. For this analysis, we could use a sub-sample, including only states/provinces where the authors previously identified the positive correlation between precipitation and the second month lag of Nino 3.4. A similar analysis to table s1 could let us identify the local effect in areas where ENSO produces wet anomalies.

Thank you for pointing out a lack of clarity in our explanation here. We begin our sample selection by defining areas of the world that are “teleconnected” with ENSO conditions based on a definition also utilised in Hsiang, Meng, and Cane (2011). This utilises the correlation through time of local temperatures and remote NINO3.4 SSTs (though the method is robust to using other NINO indices). This correlation is overwhelmingly positive, meaning that when the climate is in an El Niño phase of ENSO, local temperatures in the teleconnected regions tend to be higher. However, this is not the case for precipitation correlations, which for some locations is negative (i.e., during an El Niño phase, these regions have lower than average precipitation) and for some locations is positive (i.e., during an El Niño phase, these regions have higher than average precipitation). Of course, during a La Niña phase, these average conditions would be reversed (i.e., lower NINO3.4 SSTs would result in most teleconnected regions having lower than average temperatures, with the negatively correlated regions having more precipitation than average and the positively correlated regions having less precipitation than average).

In direct response to your point, then, we observe that locations that are shown to have a positive precipitation correlation with NINO3.4 SSTs have both wetter than average years and drier than average years during the El Niño and La Niña phases, respectively (with negative precipitation correlation locations the opposite). Therefore, places that routinely flood during either El Niño or

La Niña conditions are already included within our sample. In the main results (Table 1, figure 2b) the coefficient on the “NINO3.4 * I(Positive_Precip)” can be interpreted as restricting the effect to the subsample consisting only of those states/provinces where we identified the positive precipitation correlation.

We have updated lines 72-77 in the main text in response to this comment, and have added the following text for additional clarity in the methods section about the interpretation of the precipitation-teleconnection interaction term (line 276):

We perform a similar analysis with precipitation, defining a pixel as teleconnected if precipitation in month t is significantly correlated with the second month lag ($t - 2$) of the ENSO state (NINO3.4 SST index) for at least three months of the year. These teleconnected areas are shown in Figure 1D. This allows us to differentiate between places where warmer ENSO leads to dry or wet precipitation anomalies. We separately estimate effects of ENSO in DHS clusters located in first-level administrative units (e.g., state/province) where more than 50% of land area has three months or more per year with a statistically significant positive correlation between precipitation and the second monthly lag of NINO3.4.

Reviewer 2

We thank the referee for all the positive comments and support for the paper. They have substantially improved our paper. We address each point individually below (your comments in **bold**, ours in plain text, and excerpts from the paper indented).

This is an interesting and impressive manuscript looking at the impacts of ENSO on childhood undernutrition around the world. It is extremely important to understand the impacts of such climate oscillations on various aspects of human health, and I believe the author's research question has merit. However, when studying the impacts of climate on health it is critical to carefully consider, account for, and discuss the potential mechanisms through which changes in climate can influence the health outcome of interest. I believe the authors have done sound analysis but have not adequately addressed these mechanisms or the motivation and implications of their work. For this to be published, I believe the authors must make major revisions to the framing and discussion of the paper's research questions, methods, and results.

I have laid out specific comments below:

- 1. Line 15: "Warmer, drier El Niño conditions" is a bit misleading as it doesn't distinguish between the SST effects of ENSO and the local meteorological effects of ENSO. Throughout the paper the authors should make sure they are being careful about differentiating between SST and teleconnection effects of ENSO.**

Thank you very much for pointing out our ambiguity in language here. We have changed the text throughout the paper to make it clear when we are talking about local conditions versus the global state of ENSO. This is especially important as we find substantial heterogeneity across regions that have warmer, drier weather and regions that have warmer, wetter weather when SSTs increase to signify an El Niño phase of ENSO. We thank you for this correction, and have clarified in the sentence removing the word "drier" to make it clear that we are discussing the warmer ENSO state. The sentence now reads (line 15): "Warmer El Niño conditions predict worse child undernutrition in most of the developing world, but better outcomes in the small number of areas where precipitation is positively affected by warmer ENSO."

- 2. Line 29-35: I'm unfortunately not convinced by the arguments made here to motivate the analysis. The authors state that "much of humanity is still susceptible to its consequences" when the reality is that we don't yet understand most of these consequences. More importantly, the authors argue that most existing studies focus on single countries and that larger scale studies are needed. However, I would argue that local studies are actually more important for informing context-specific effects**

and interventions. The relationship between ENSO and any outcome is highly dependent on the impact that ENSO has on local meteorology as well as local infrastructure, wealth, and development. Can the authors expand more on (1) What is the benefit of zooming out spatially? (2) Are there really going to be policies or interventions at the global or multi-national scale? If so, what are these and provide some examples. (3) Is it valid to study a climate phenomenon on such a large spatial scale that has such specific local effects?

We thank the referee for the opportunity to better outline the contributions of this work. We take the questions in turn. First, there are multiple benefits to an analysis that zooms out spatially as a complement to analyses on single locations. Global-scale phenomena such as ENSO affect multiple countries, resulting in general equilibrium effects operating at regional or global level through changes in food or energy prices and in international trade, for example. Financial transfers may occur from one country to another through foreign aid or remittances. An analysis on a single country may miss larger-scale consequences that are only observed and properly quantified through global analysis. It is for this reason that so much of the literature on climate impacts takes a global perspective, as mentioned below.

With regards to the second question: there are many policies, interventions, and institutions that operate at regional and global scale. These include regional development banks (for example, the Inter-American Development Bank or the African Development Bank) as well as international institutions (such as UN agencies like UNICEF or the World Food Programme, the 2020 Nobel Peace Prize Laureate) and international non-profits that respond to disasters (such as the International Red Cross). All of these institutions are involved in activities supporting countries with technical and financial support to enhance resilience to environmental disasters, to recover from those disasters, and to engage in humanitarian support. In addition, the high-income OECD nations engage in bilateral support to developing countries. A criticism of the multiple actors involved in these activities is the lack of coordination, and one of the contributions of systematic documentation of the effect of a predictable phenomenon like ENSO is that it allows for improved response time and coordination across multiple development agencies in anticipation of ENSO-related climatic shocks and consequences.

The third question asks whether it is valid to study a climate phenomenon at large spatial scale when the effects are locally specific. We note first that our methodological approach stems from the literature on econometric climate change impacts, a relatively new and rapidly advancing interdisciplinary field which has developed a body of statistical techniques for evaluating relationships between climatic and social outcomes, heavily motivated by understanding the future effects of climate change. We rely substantially on the methodological approaches

developed by this field, including many papers that are global in scope.^{5,6,7,8,9,10} Many of the effects of climate on human outcomes -- from agriculture to conflict to income levels to mortality -- surely are mediated by local context and population characteristics that determine the exact vulnerability people experience in the face of climate shocks. Nevertheless, the literature has placed high value on understanding these relationships at large scale in order to see the generalizability of basic relationships and to use the power of large datasets to explore variations in the relationships across different contexts. A final point is that large datasets that can be brought to bear in global analyses provide statistical power that allows estimating causal effects using econometric specifications that would likely be underpowered using smaller datasets at national level.

Finally, we have removed the clause in the introduction "...much of humanity is still susceptible to its consequences", agreeing with the referee that instead of stating this as fact, this paper seeks to quantify to what degree large parts of the developing world are susceptible to ENSO's effects.

3. Can the authors include more information in the introduction about possible mechanisms explaining the connection between ENSO and malnutrition? Why would we expect to see these effects and what have existing studies found related to their study question? Further, and related to the above comment, how are the authors adding to the existing literature and enhancing our understanding of the mechanistic impacts of ENSO on childhood malnutrition?

We thank the referee for the opportunity to clarify the mechanisms through which ENSO affects human health. We have added the following paragraph to detail the mechanisms documented in the literature and our contribution estimating the total effect of ENSO on child nutrition (line 47):

ENSO has destabilizing effects on agriculture (Iizumi et al 2014, David 2002), economic production (Hsiang and Meng 2015), and social stability (Hsiang, Meng and Cane 2011) throughout teleconnected areas of the global tropics. It has been linked to human health outcomes directly through its effects on vector- and water-borne

⁵ Iizumi, T., J.-J. Luo, A.J. Challinor, G. Sakurai, M. Yokozawa, H. Sakuma, M.E. Brown and T. Yamagata, "Impacts of El Niño Southern Oscillation on the global yields of major crops," *Nature Communications* 5, no. 3712, (2014).

⁶ Hsiang, Solomon M., Marshall Burke, and Edward Miguel. "Quantifying the influence of climate on human conflict." *Science* 341.6151 (2013): 1235367.

⁷ Dell, Melissa, Benjamin F. Jones, and Benjamin A. Olken. 2014. "What do We Learn from the Weather? The New Climate-Economy Literature." *Journal of Economic Literature* 52 (3): 740-98.

⁸ Dell, Melissa, Benjamin F. Jones, and Benjamin A. Olken. 2012. "Temperature Shocks and Economic Growth: Evidence from the last Half Century." *American Economic Journal: Macroeconomics* 4(3): 66-95.

⁹ Burke, M, SM. Hsiang, and E Miguel. "Global non-linear effect of temperature on economic production." *Nature* 527.7577 (2015): 235-239.

¹⁰ Hsiang, S., K. Meng and M. Cane. "Civil conflicts are associated with the global climate," *Nature*, 476, pp. 438-441. 2011.

infectious diseases (Thomson et al. 2005, WHO 1999, Cazelles et al 2005, Pascual et al 2000, Bennet et al 2012), and indirectly by decreasing agricultural yields and increasing food insecurity (Rosenzweig and Hillel 2008) as well as the likelihood of conflict (Hsiang Meng and Cane 2011). ENSO's adverse effects on yields are particularly acute in the tropics (Iizumi et al 2014, Hsiang and Meng 2015), where the vulnerable population of food insecure children is larger and temperatures are closer to critical crop collapse thresholds (Schlenker and Roberts 2009, Schlenker and Lobell 2010). Our interest is thus in the total influence of ENSO variability through all plausible mechanisms – from agricultural productivity to infectious disease to conflict – that are known to affect human nutrition, as well as the systematic differences in ENSO response across places with different precipitation responses to ENSO, across continents, and across decades.

We have clarified in the paper that our aim is to measure the total effect of ENSO on child nutrition through *all* potential mechanisms, and to do so for *all* affected countries with data, as this has not been accomplished to date in the literature. Disentangling and parameterizing the multiple different causal chains at different scales would be a momentous undertaking and beyond the scope of a single paper; indeed, we plan to build on these findings and explore specific mechanisms and their relationships to global ENSO state in future work. Regardless, measuring the total effect of ENSO on child nutrition is relevant for public policy, an issue we now speak to in the final paragraph of the manuscript.

We have also followed the referee's advice and expanded our discussion of our contribution to understanding the mechanism behind ENSO's effects. This new text reads (line 203):

The consistency across regions suggests that local context is not generating first-order differences in how ENSO affects children's health, with the exception of whether the NINO3.4 measure is positively or negatively correlated to local precipitation. The importance of precipitation correlation in determining ENSO's effect suggests that agriculture plays a strong role in linking ENSO to nutrition outcomes. Other potential channels, such as vector-borne or water-borne diseases, would likely increase with higher precipitation, as would impacts of flooding. Since these other channels are not translating into adverse health outcomes for children in places where precipitation increases during the El Nino state, the improved nutrition observed in the data in these locations suggests that increases in agricultural production due to higher rainfall are driving the health effect. Similarly, in places where precipitation decreases during the El Nino state, the adverse health outcomes for children are consistent with decreased agricultural production and are not consistent with a decrease in water-borne or vector-borne diseases that may be expected during a reduced rainfall year. Finally, the fact that the effect of a warmer ENSO state varies by the direction of precipitation

correlation suggests that the health effects are primarily operating through the precipitation channel and not the increase in temperature, since all places experience a temperature increase.

4. Line 45: I'm not sure what the authors mean by "documenting systematic differences in ENSO response within sample."

Thank you for pointing this out. We have clarified in the text what forms of heterogeneity in ENSO's effects we examine, specifically: heterogeneity due to different relationships between NINO3.4 SSTs and the direction of the effect on local precipitation, heterogeneity across broad (continental) areas in our sample; and heterogeneity through time by looking at effects in different decade.

The heterogeneity in precipitation is particularly important if the mechanism of our estimated effect goes through agriculture, and this leads us to two distinct choices in our research design. First, our main results throughout the paper estimate treatment effects for the positive and negatively correlated locations separately via an interaction term. We begin our sample selection by defining areas of the world that are "teleconnected" with ENSO conditions based on a definition also utilised in Hsiang, Meng, and Cane (2011). This utilises the correlation through time of local temperatures and remote NINO3.4 SSTs (though the method is robust to using other NINO indices). This correlation is primarily positive (see Figure 1C), meaning that when the climate is in an El Niño phase of ENSO, local temperatures in the teleconnected regions tend to be higher. However, this is not the case for precipitation correlations, which for some locations is negative (i.e., during an El Niño phase, these regions have lower than average precipitation) and for some locations is positive (i.e., during an El Niño phase, these regions have higher than average precipitation). Of course, during a La Niña phase, these average conditions would be reversed (i.e., lower NINO3.4 SSTs would result in most teleconnected regions having lower than average temperatures, with the negatively correlated regions having more precipitation than average and the positively correlated regions having less precipitation than average). This systematic difference in precipitation responses forms a central part of our analysis. Other forms of heterogeneity are shown as robustness, and we find that both across regions of the world and through time, the effects do not statistically differ from our main estimates.

Again, thank you for pointing out the lack of clarity, and we have clarified in the text as follows (line 55): "...as well as the systematic differences in ENSO effects across places with different precipitation responses to ENSO, across continents, and across decades."

5. Why did the authors choose to use Niño 3.4 Index as opposed to other indices (Niño 4, SOI, MEI, ONI)? Different regions of the world have different levels of sensitivity

to different ENSO indices. It would be worth doing some sensitivity analyses with other indices to see how the results differ.

Thank you for this comment, as it has substantially improved the robustness of our analysis. We specifically choose NINO3.4 based on the findings in Barnston et al. (1997)¹¹, which identified the NINO3.4 region as the most appropriate for understanding ENSO and ENSO-related teleconnections over land areas. In that paper, the authors note that the original four NINO indices (NINO1 through NINO4) were:

selected partly for convenience with respect to the locations of the SST data-producing ship tracks. Nino 1 and Nino 2 reflect the South American coastal SST where ENSO has great economic impacts. Nino 3 and Nino 4 are larger regions located in the eastern and central equatorial Pacific, respectively. Monthly SST anomalies in each of the Nino areas have been monitored in real-time since 1982. In this note we inquire whether these four regions might have been defined differently if we had known in the early 1980s what we now (in 1997) know about ENSO.

In particular, the authors note that NINO3.4 is much better suited to analysis of the ENSO-related phenomena that affects climate over land rather than conditions in the ocean.

With that in mind, we conduct an analysis for teleconnections definitions using the NINO3 and NINO4 indices. This follows the analysis in the text for identifying teleconnected areas, and that description has been expanded in response to earlier comments. Following Hsiang, Meng, and Cane (*Nature*, 2011), a pixel is considered teleconnected if it has at least three months of the year when its temperature is statistically significantly correlated with the second lag of the NINO SST index value. For the current comparison, we conduct the analysis using the ERA-Interim reanalysis data. This was not the preferred climate dataset in the main analysis, as it has a shorter time series than the University of Delaware climate dataset (UDEL). We also preferred to use a gridded station dataset for our main analysis rather than a reanalysis. However, station-measurements data are not available over the ocean where much of the difference between teleconnection is seen. Therefore, we use ERA data for this comparison so the differences in the ocean are clear.

These results are presented in an updated version of Figure S1 in the paper, which is reproduced below. The colors represent the number of months that exhibit a significant correlation for each pixel, with darker colors signifying that more of the year's climate in that location is affected by ENSO. Clearly, the number of correlated months in the equatorial Pacific is high by construction. The three upper panels all show the characteristic ENSO pattern of teleconnection.

¹¹ Bamston, Anthony G., Muthuvel Chelliah, and Stanley B. Goldenberg. "Documentation of a highly ENSO-related SST region in the equatorial Pacific: Research note." *Atmosphere-ocean* 35, no. 3 (1997): 367-383.

We see little difference between the NINO3.4 teleconnection with ERA and with UDEL. Across the top three panels, using NINO3, NINO4, and NINO3.4 SST time series, we see virtually no difference in the areas of teleconnection over land, with some minor differences in locations and intensity over the oceans. For this reason, we proceed with NINO3.4 as the main SST index for our analysis. We have updated the text to reflect these results (line 272): “We note that other NINO indices were tested for this analysis, and teleconnected land areas are largely unchanged, as shown in Figure S1. These teleconnected areas are shown in Figure 1C.”

Figure S1: Comparison of teleconnection patterns using alternate NINO SST indices. Colors indicate the number of months for which a pixel’s temperature (measured using the ERA-Interim reanalysis) exhibits a statistically significant correlation with the respective SST index.

6. Line 57: The authors should further explain the spring barrier delay, why it is important to account for, and how their methods address it.

Thank you for this comment, which we have tried to address by clarifying in the text. The spring barrier¹² is a feature of the organization of ENSO. As discussed¹³ by NOAA, “the spring barrier is said to exist because spring is a transitional time of year for ENSO. The spring is when ENSO is shifting around— often El Niño/La Niña events are decaying after their winter peak, sometimes passing through Neutral, before sometimes leading to El Niño/La Niña later on in the year.” In chapter 2 of their volume on ENSO, Sarachik and Cane (2010) note that “there is some variation in the evolution of the warm phases, but by and large, the warm phases grow during the spring of year (0), peak towards the end of year (0) and decay during the spring of year (+1).”

While the physical explanation of the oceanographic features of ENSO are outside the scope of our paper, we have amended the text to explain why it is important to account for and address the spring barrier. We have inserted the following text (line 296):

ENSO events typically begin to develop in late northern hemisphere spring, peak at the year’s end, and decay during the early spring of the following year. For this reason, the calendar year is not aligned with exposure to this phenomenon, and specifying exposure that occurred in the early spring of one year would lead to mis-assignment of treatment and biased estimates. We therefore match survey timing from May of year t to April of year $t+1$ to an ENSO exposure that captures the mean El Niño or La Niña magnitude in a year (occurring between May and December). This ensures that the specific timing of ENSO events is accounted for in our analysis.

7. I’m a little concerned about the annual resolution of the malnutrition outcome. Often teleconnections are felt most strongly in specific seasons, and these timings differ in different countries. Is it possible that there are children’s weight measurements that were taken BEFORE any effect of ENSO is felt locally? Would it be possible to stratify the outcome by date of measurement to see if there are stronger effects of ENSO on malnutrition at certain times of the year?

We thank the referee for raising this concern. It is correct to point out that we have not dealt substantially with heterogeneity *within* a single ENSO “episode” as it is occurring. While interesting, this is beyond the concerns of the current paper, and would do little to change the interpretation. The first point to make regarding the referee’s concern -- that a child might have been surveyed by the DHS before the onset of ENSO’s effects -- becomes a matter of treatment misassignment in the econometric specification (a child being coded as “treated” by an ENSO

¹² Sarachik, Edward S., and Mark A. Cane. *The El Niño-southern oscillation phenomenon*. Cambridge University Press, 2010.

¹³ <https://www.climate.gov/news-features/blogs/enso/spring-predictability-barrier-we%E2%80%99d-rather-be-spring-break>

state despite being surveyed while the consequences of that ENSO state have not manifested). This misassignment would lead to classical measurement error and hence an underestimation of ENSO's effect on child outcomes, which would mean that our estimates are not invalid but instead can be interpreted as lower bounds.

We point out one major benefit of defining exposure over an extended period, which is to account for varying lagged effects of a single ENSO event, even ones that differ across space. This occurs mechanically, as the following example will suggest: if country X experiences effects on local temperatures with a 2 month lag, and country Y experiences effects with a 3 month lag, as long as our ENSO variable is a good proxy for overall ENSO intensity, we are still measuring the effect of an ENSO event in both country X and Y as this measure captures the total effect of that episode in both regions. We address further concerns raised by this comment in a number of ways.

First, through our teleconnection definition we identify those areas whose climates are affected by ENSO in a broad sense. Our maps of teleconnections allay fears that we are being overly conservative in the assignment of teleconnection, conditional on there being household survey data within a country. Each of these countries may be affected by ENSO in a particular way, which, as the referee points out, may vary before, during, and after the peak. We identify the aggregate effect of annual ENSO states as they manifest through various channels and timescales to yield the average population outcomes we document, and contend that the annual state is of policy importance. This becomes particularly salient given the body of evidence (outlined in response to comment 3, above) that ENSO influences trade, financial markets, and agricultural outcomes both across and within countries, complicating the timing and spatial distribution of effects.

Second, the comment suggests that there may be different outcomes that occur at different points within the cycle of a single ENSO event. To understand the extent to which nutritional outcomes may be affected differently if they are measured at different points within the ENSO cycle, the graph below shows results of estimating our main effect interacted with the season of measurement. The results in the figure (now added to the SI as Figure S3), show the effect for the main (non-positively precipitation-correlated) sample, which provides sufficient sample size and variation to explore intraannual heterogeneity. We observe three things: i) the standard errors increase because each coefficient shown below is identified using approximately one quarter the sample size in the main regression, ii) none of the seasonal coefficients are statistically different from our main effect, and iii) an F-test fails to reject that the smallest (SON) and the largest (JJA) effects shown here are the same ($p=0.25$). What this implies is that the effects on nutrition of an ENSO episode (typically peaking in Nov, Dec, Jan) do not vary by season in which a person is measured, whether that is before the peak (i.e., JJA, SON), during the peak (i.e., DJF), or after the peak (i.e., MAM).

8. Line 62: This sentence needs citations, or the authors should make it clearer that this is an original analysis.

Thank you for this comment. We have made it clear that this is an original analysis, building on work in Hsiang, Meng, and Cane (*Nature*, 2011). In particular, the methods section now describes in much more detail the teleconnection analysis performed for this paper, including the novel precipitation analysis (see description on lines 264-292 and SI line 117).

9. Line 67: Can the authors discuss what the possible confounders are that they are trying to control? This gets back to the mechanism question and what causal pathway they are trying to isolate. Also, why are the authors removing monthly seasonality by major world regions instead of by country? Generally, the structure of the models used need to be discussed more in terms of the mechanisms they are trying to estimate.

We apologize for being overly succinct on this topic in the previous draft. We now explicitly note both the spatiotemporal as well as individual-level controls in the introduction, and have substantially increased discussion of mechanisms (as outlined in responses to comments 3 and 13), including a new paragraph (line 79):

A key aspect of our research design rests on exploiting the temporal variability of the ENSO cycle. While ENSO follows a variety of non-random patterns, such as the general progression from El-Niño state to La-Niña state, the timing of event occurrence is sufficiently stochastic that even state of the art models have limited prediction skill beyond six months into the future (Tang et al. 2018). We thus use variation in ENSO anomalies – measured as a deviation from long-run average conditions – in order to statistically isolate the effect of variation in ENSO state on child malnutrition. Following standard practice in the climate impacts literature (Dell Jones and Olken 2014, Hsiang 2016), we purge the estimates of potentially confounding average differences between countries and within them based on rural and urban areas using fixed effects (indicator variables) for spatial location, detrend the data by major world regions using an annual trend, remove monthly seasonality by major world regions using month fixed effects, and include simple country-specific controls for mother's age at child's birth and total years of mother's education. Our results correspond to comparing children surveyed at different times in the same country but under different ENSO conditions, while appropriately detrending the data and controlling for average health differences across countries and regions.

To address the question about seasonality, we aggregate month fixed effects up to the regional scale to preserve identifying variation in the ENSO cycle. Because the DHS are collected with idiosyncratic timing within the year, there are often months within a country that are only observed once. For example, if DHS were collected April-Sept in one country's survey wave, and July-Dec in the same country's second wave a few years later, then April, May, and June as well as Oct, Nov, and Dec would each only show up once in the data. This means that a country-by-month fixed effect would be collinear with ENSO state for children measured in those singly-observed months, since all children observed within that month get the same annual ENSO value, meaning they do not contribute any identifying variation in ENSO to the main regression. Regardless, the country-by-month fixed effect specification does look very similar to our main result, albeit noisier, as indicated in the SI Table S1 column 5. We now expand on this point further in the discussion of Table S1 in the SI (line 22):

Model (5) adds country-specific interview month fixed effects to remove seasonality separately by country, rather than by the larger UNICEF region. Because the DHS are collected with idiosyncratic timing within the year, there are often months within a country that are only observed once. This means that a country-by-month fixed effect would be collinear with ENSO state for children measured in those singly-observed months, since all children observed within that month get the same annual ENSO value, meaning they do not contribute any identifying variation in ENSO to the main

regression. Regardless, while this stricter specification makes estimates slightly noisier, they are not statistically different from those in (4).

10. Line 91: What do the authors mean by “consistent with higher noise in height measurements, especially for very sick children”? Can they elaborate on this?

Height is more noisily measured than weight for a variety of reasons, including that height / length measurements are often read off physical rulers rather than directly observed using a digital scale as with weight, that child posture can affect height measurements but not weight, and that particularly young or sick children may be difficult to position and measure accurately per the DHS. This has been noted to be the case by both the WHO and the Demographic and Health Surveys Program¹⁴. Because height is the denominator in the weight for height ratio, measurement error is particularly pernicious, leading to geometric / proportional errors rather than linear ones, as would happen if it were the numerator. This produces larger error variance and hence larger standard errors. We now cite the DHS anthropometrics methods report and explain this in more detail in the main text (line 109), saying “consistent with higher measurement error in height measurements due to the difficulty of measuring child height / length compared to weight (Pullum et al., 2020), which decreases the precision of our estimates.”

11. Line 95: Please expand on this sentence about agriculture. Again, it is important to fully discuss the mechanistic connections and implications of the results.

We’ve attempted to follow this comment as closely as possible, and this has led us to remove the sentence in question from the results section, place it into the discussion section, and expand on the interpretation as requested. The text now reads (line 203):

The consistency across regions suggests that local context is not generating first-order differences in how ENSO affects children’s health, with the exception of whether the NINO3.4 measure is positively or negatively correlated to local precipitation. The importance of precipitation correlation in determining ENSO’s effect suggests that agriculture plays a strong role in linking ENSO to nutrition outcomes. Other potential channels, such as vector-borne or water-borne diseases, would likely increase with higher precipitation, as would impacts of flooding. Since these other channels are not translating into adverse health outcomes for children in places where precipitation increases during the El Niño state, the improved nutrition observed in the data in these

¹⁴ See, e.g., WHO guidelines https://www.who.int/childgrowth/publications/physical_status/en/ and in particular the DHS documentation on error sources in anthropometrics: <https://dhsprogram.com/publications/publication-MR28-Methodological-Reports.cfm>

locations suggests that increases in agricultural production due to higher rainfall are driving the health effect. Similarly, in places where precipitation decreases during the El Niño state, the adverse health outcomes for children are consistent with decreased agricultural production and are not consistent with a decrease in water-borne or vector-borne diseases that may be expected during a reduced rainfall year. Finally, the fact that the effect of a warmer ENSO state varies by the direction of precipitation correlation suggests that the health effects are primarily operating through the precipitation channel and not the increase in temperature, since all places experience a temperature increase.

12. Line 98: I would recommend removing this analysis from the paper. The impact of Interventions on malnutrition are extremely dependent on local contexts, so I'm not convinced by the authors applying existing effect estimates from one paper to the entire region of study. I don't feel that this analysis adds useful or realistic information to the paper and the manuscript would be stronger without it.

This calculation is provided as a reasonable back-of-envelope estimate in order to provide the reader with a sense of the magnitude and importance of the effect of ENSO. In fact, *Nature* editor Michael White told us when we were preparing this manuscript that any paper measuring the nutrition impacts of ENSO should benchmark the effects against other documented public health effects on nutrition. While the regressions estimate ENSO's effect on the average child in the countries we study, that estimate does not communicate the aggregate scale of the impact over hundreds of millions of children. We contend that benchmarking these effects against documented public health interventions serves to not only remind the reader of the massive human toll that we are estimating, but also of the magnitude of the costs that would be required to offset these effects. It is also worth noting that the numbers drawn from Bhutta et al. are calculated using a meta-analysis that is explicitly directed at the idea that local context matters. The numbers in Bhutta et al. (*Lancet*, 2013) that we use come from studies from Ghana, India, Ecuador, Nigeria, South Africa, Vietnam, Pakistan, China, Bangladesh (reviewed by ref. 54 in Bhutta et al.) plus Cambodia, Peru, Indonesia, Mexico, Philippines, Algeria, Botswana, Tanzania (reviewed by ref. 58 in Bhutta et al.). The heterogeneity across these regions is captured by the confidence intervals in the meta-analyses, which we use to make our confidence intervals in Figure 3. Therefore we are capturing both the global average response and the dependence on local contexts in our comparison. An important note in response to the idea that local context renders such a meta-analysis as that performed in Bhutta et al. uninformative is that each of the interventions have unambiguously positive effects that are significantly different than zero.

In order to better motivate this calculation for the reader, we have moved the following sentence earlier in the paragraph (line 118): "To give context to the size of these effects, we look at the scale of public health interventions needed to offset an event of similar magnitude to the 2015 El Niño, using published effect sizes of nutritional interventions."

13. Discussion: This section needs much more discussion of the possible mechanisms connecting ENSO to malnutrition that would explain the authors' results.

We appreciate the invitation to expand the discussion of possible mechanisms. As explained in the response to comment 3 above, we have added a paragraph on mechanisms to the main text (line 47):

ENSO has destabilizing effects on agriculture,^{15,16} economic production¹⁷, and social stability¹⁸ throughout teleconnected areas of the global tropics, and has been linked to human health outcomes directly through its effects on vector- and water-borne infectious diseases^{19,20,21,22,23}, and indirectly through its effects on agricultural output²⁴ and likelihood of conflict²⁵. ENSO's adverse effects on yields are particularly acute in the tropics²⁶, which is where the vulnerable population of food insecure children is larger. Our interest is in the total effect of ENSO variability across all channels of influence – from agricultural productivity to infectious disease to conflict – that might affect human nutrition.

In addition, we have followed the referee's advice and reworked the discussion to go deeper into mechanisms that have been documented, and highlight two points about mechanisms that our results suggest (line 199):

¹⁵ Iizumi, T., J.-J. Luo, A.J. Challinor, G. Sakurai, M. Yokozawa, H. Sakuma, M.E. Brown and T. Yamagata, "Impacts of El Niño Southern Oscillation on the global yields of major crops," *Nature Communications* **5**, no. 3712, (2014).

¹⁶ David, M., "Late Victorian holocausts: El Niño famines and the making of the third world," London: Verso, 2002.

¹⁷ Hsiang, S. M. and Meng, K.C. "Tropical Economics." *American Economic Review: Papers & Proceedings* 2015, 105(5): 257-261.

¹⁸ Hsiang, S., K. Meng and M. Cane. "Civil conflicts are associated with the global climate," *Nature*, vol. 476, pp. 438-441. 2011.

¹⁹ M. Thomson, S. Mason, T. Phindela and S. Connor, "Use of rainfall and sea surface temperature monitoring for malaria early warning in Botswana," *Am. J. Trop. Med. Hyg.*, vol. 73, no. 1, pp. 214-221, 2005.

²⁰ World Health Organization, "El Niño and Health," WHO, Geneva, 1999.

²¹ B. Cazelles, M. Chavez, A. McMichael and S. Hales, "Nonstationary influence of El Niño on the synchronous dengue epidemics in Thailand," *PLoS Med*, vol. 2, no. e106, 2005.

²² M. Pascual, X. Rodo, S. Ellner, R. Colwell and M. Bouma, "Cholera dynamics and El Niño-Southern Oscillation," *Science*, vol. 289, pp. 1766-1769, 2000.

²³ A. Bennett, L. Epstein, R. Gilman, V. Cama, C. Bern, L. Cabrera, A. Lescano, J. Patz, C. Carcamo, C. Sterling and W. Checkley, "Effects of the 1997-1998 El Niño episode on community rates of diarrhea," *Am J Public Health*, vol. 102, pp. e63-69, 2012.

²⁴ C. Rosenzweig and D. Hillel, *Climate Variability and the Global Harvest: Impact of El Niño and Other Oscillations of Agroecosystems*, Oxford Univ. Press, 2008.

²⁵ Hsiang, S., K. Meng and M. Cane. "Civil conflicts are associated with the global climate," *Nature*, **476**, pp. 438-441. (2011).

²⁶ Hsiang, S. M. and Meng, K.C. "Tropical Economics." *American Economic Review: Papers & Proceedings* 2015, 105(5): 257-261.

This analysis measures the total effect of ENSO on child nutrition through all potential measurements and for all affected countries with available data. The negative relationship between child nutrition and warm ENSO state does not appear to vary appreciably across space, with the effects for major world regions in the sample being statistically indistinguishable from our main effect (Fig 2C). The consistency across regions suggests that local context is not generating first-order differences in how ENSO affects children's health, with the exception of whether the NINO3.4 measure is positively or negatively correlated to local precipitation. The importance of precipitation correlation in determining ENSO's effect suggests that agriculture plays a strong role in linking ENSO to nutrition outcomes. Other potential channels, such as vector-borne or water-borne diseases, would likely increase with higher precipitation, as would impacts of flooding. Since these other channels are not translating into adverse health outcomes for children in places where precipitation increases during the El Niño state, the improved nutrition observed in the data in these locations suggests that increases in agricultural production due to higher rainfall are driving the health effect. Similarly, in places where precipitation decreases during the El Niño state, the adverse health outcomes for children are consistent with decreased agricultural production and are not consistent with a decrease in water-borne or vector-borne diseases that may be expected during a reduced rainfall year. Finally, the fact that the effect of a warmer ENSO state varies by the direction of precipitation correlation suggests that the health effects are primarily operating through the precipitation channel and not the increase in temperature, since all places experience a temperature increase.

14. Additionally, the authors should discuss in more depth how the results could be used to inform specific interventions.

An important part of this work places the effect sizes in context by comparing them with effect sizes from major nutritional interventions.²⁷ We find that a 2015 El Niño-type event could set back progress towards the 2030 Sustainable Development Goals (SDGs) nutritional goal by a full year, and that between 72 million and 134 million children would need to be targeted with nutritional interventions to offset its effects. Our methods could thus be paired with existing ENSO forecasts to form the basis of a "starvation early warning" system, yielding immediate practical applications beyond characterization of the relationship. In 2020, for example, the World Food Programme warned of one of the most extensive famines in history, affecting hundreds of millions of people, due to the unexpected economic effects of the Covid-19 crisis. Last summer, the UN's outgoing special rapporteur on extreme poverty and human rights stated

²⁷ Bhutta, Z.A., Das, J.K., Rizvi, A., Gaffey, M.F., Walker, N., Horton, S., Webb, P., Lartey, A., Black, R.E., T.L.N.I.R. Group & Maternal and Child Nutrition Study Group. Evidence-based interventions for improvement of maternal and child nutrition: what can be done and at what cost? *The Lancet* 382, 452-77 (2013).

that we are completely off track to meet the 2030 SDGs. Our work identifies a component of global malnutrition that society can predict, and we believe it is of significant general interest to both scientists and policy-makers as they work to minimize risk of starvation and food insecurity.

Following the referee's advice, we have expanded our discussion of these points in the concluding paragraph of the paper (line 236):

Hunger continues to affect hundreds of millions of people, with shocks like COVID-19 capable of generating extensive famine and suffering. Our work identifies a component of global undernutrition that society can predict, and can contribute to a hunger early warning system that allows actors to deploy nutrition and humanitarian support operations in proactive instead of reactive ways. Governments and agencies engaged in multi-year humanitarian planning and budgetary frameworks could incorporate ENSO forecasts to anticipate fluctuations in availability of local vs. global resource streams needed to ensure progress in fighting malnutrition. Despite scientific progress characterizing ENSO and documenting its various channels of influence on society, much remains to be done to decouple nutrition outcomes in developing countries from this predictable interannual phenomenon.

15. Data: Why are the authors using the maximum monthly Niño 3.4 anomaly value from May - Dec? Can they provide a justification for this choice?

Thank you for pointing out that our draft lacked clarity on this point. We particularly appreciate your asking this, as it alerted us to a labelling error in our draft, where the main model was previously presented as using the “max” ENSO as opposed to the “mean” ENSO. We apologise for this and have corrected the labelling on the main estimates to be the effect of the *mean* of the ENSO index from May to December. The reason to choose May as the starting point is due to the seasonality in ENSO, which tends to develop in late spring, peak at the end of the calendar year, and decay in the spring of the following year. Our reason to choose the December cutoff is largely to maintain consistency and comparability with other papers in the literature (e.g., Hsiang, Meng, & Cane, 2011). However, the choice of a specific summary statistic and the conditions it must satisfy are largely empirically driven. As noted in previous comments, we do not wish to examine the sub-seasonal heterogeneity of these results. Therefore, our ENSO exposure measure must accurately capture the nature and state of ENSO in a given year. As in the original draft, we explore robustness to the choice of averaging period in Table S3. Looking at the main coefficient of interest in that table reveals that the values for the May-April max (-0.0301 [95% CI: -0.0588 - -0.00137]), the May-Dec max (-0.0306 [95% CI: -0.0601 - -0.0012]), the May-April mean (-0.0215 [95% CI: -0.0425 - -0.0005]), and the May-April max of the 3-

month moving average (-0.0322 [95% CI: -0.0631 - -0.0013]) are all within the confidence interval our main result, the May-Dec mean (-0.0251 [95% CI: -0.0466 - -0.0036]). Moreover, the choice of this exposure measure leaves our qualitative and quantitative conclusions unchanged. We have added a more detailed description of this supplemental table to the main text (line 143):

Alternative indicators for ENSO state yield similar results (Table S3), with positive deviations from the mean ENSO state decreasing anthropometric z-scores, and negative deviations from the mean ENSO state (La Niña events) reducing undernutrition (opposite patterns occur in the few places where precipitation increases with ENSO SST). Coefficients for alternate definitions of ENSO are not statistically distinguishable from our main effect.

16. Line 168: How did the authors come up with this method for defining teleconnections? It should be motivated and if possible, citations should be provided to support the use of this method. Also, it is unclear as written if the authors did the teleconnection analysis themselves or relied on previously calculated estimates. Relatedly, why are the authors defining teleconnection based only on temperature? There are countries that have precipitation teleconnections but not temperature teleconnections.

We apologise for not making this clearer in the main text, and note that much of the discussion of this definition was contained in the previously submitted supplement. Based on these comments, we have added an expanded discussion in the methods section of the paper (starting on line 264).

The analysis on teleconnections was conducted entirely by the authors, though it follows a peer-reviewed methodology. More details are discussed below, but we first point out that there are two ways that we use teleconnections in the analysis. First, we use the teleconnection results to choose a sample. There are two ways to do this, as pointed out in your comment, by using either local precipitation or local temperature. It is for this reason, while using temperature-derived teleconnections in our main result, and based on similar reasoning as outlined in your comment, we also included results in Table S7-8 of the previously submitted draft, now Table S9-10 of the revised draft. These tables show the estimates if the sample is chosen as one where countries are defined as teleconnected if they exhibit a relationship to ENSO through either temperature or precipitation (Table S7), and one where countries are considered teleconnected if they exhibit relationships through both temperature and precipitation (Table S8). In terms of sample size, the precipitation-OR-temperature sample is the largest, followed by the temperature only and then

the precipitation-AND-temperature samples. Results are robust to any of these definitions, and maps of locations can now be seen in Figure 1.

The method for establishing teleconnected areas follows that in Hsiang, Meng, and Cane (*Nature*, 2011), which is citation number 8 in the prior draft. Notably, Mark Cane, one of the foremost scientists studying ENSO, is among the authors of that paper. This method utilises the correlation through time of local temperatures and remote NINO3.4 SSTs (though the method is robust to using other NINO indices). This correlation is positive in the vast majority of cases, meaning that when the climate is in an El Niño phase of ENSO, local temperatures in the teleconnected regions tend to be higher. However, this is not the case for precipitation correlations, which for some locations is negative (i.e., during an El Niño phase, these regions have lower than average precipitation) and for some locations is positive (i.e., during an El Niño phase, these regions have higher than average precipitation). Of course, during a La Niña phase, these average conditions would be reversed (i.e., lower NINO3.4 SSTs would result in most teleconnected regions having lower than average temperatures, with the negatively precipitation-correlated regions having more precipitation than average and the positively precipitation-correlated regions having less precipitation than average).

This heterogeneity in precipitation, a well-known consequence of ENSO, is particularly important if the mechanism of our estimated effect goes through agriculture, and motivates our second way of using the teleconnection analyses. Within the sample of places that have DHS data available, we find that 94% of them have either a negative or neutral precipitation relationship with ENSO and 6% have a positive relationship. This heterogeneity is a central feature of our main estimates.

We have changed the Methods section in the main text to include clarifying detail on this choice of sample and approach to identifying teleconnected areas (line 264).

17. The specific countries included in the analysis are not stated or shown in the main text. They must be indicated very early in the manuscript so the readers are contextually oriented to the analysis. It would be useful to see the countries that they included in Figures 1C and 1D. Figure 1D should include all countries in the study to demonstrate that most countries have negative precipitation teleconnections

Thank you for this excellent suggestion, which we follow as closely as possible. We have made the following changes to Figure 1. Both 1C and 1D now explicitly show the sample of countries that we use in estimating our main effect, via a thick black border around those countries. All countries and survey years are listed in Table S11 of the revised draft (S9 of the previous draft). Figure 1C now shows the significant pixel-level correlations with NINO3.4 SSTs and the direction of that correlation. This demonstrates the largely spatially contiguous nature of the

temperature teleconnection as well as the fact that it is almost exclusively positive. As suggested, figure 1D is now a “zoomed out” version of the previous Figure 1D showing the whole world. This displays the heterogeneity in precipitation responses that is central to our estimates, and also that the spatial extent of precipitation teleconnections is less than that of teleconnections calculated using temperature. In this way we hope we have clarified the two main components of ENSO effects that have been included in our analysis. Based on prior comments, we have also expanded the discussion of the role of our teleconnection analyses in the methods section of the revised draft (starting on line 264).

18. Figure 1C needs to show the direction of the association as well as the strength of the association — Why have the authors chosen to use the number of months with associations?

Thank you for pointing out the lack of clarity here. We have changed the figure in response to this and prior comments. In terms of choosing numbers of months with associations, we follow Hsiang, Meng, and Cane (*Nature*, 2011) in this analysis. We have clarified this in the methods section of the main text (line 264).

19. Figure 2C: Can the authors specify in the legend which models produced these estimates?

We have now done so for all models in Figure 2.

20. Supplementary Material: The authors include many sensitivity analyses but it is unclear why they are performing each analysis, especially as many are not discussed in the main text. Each analysis should be motivated and linked to the hypothesized mechanisms and causal pathways.

We agree with the need for more explanation raised both here and in prior comments, and have considerably expanded the discussion of sensitivity analyses in the Results section of the main text (line 137-183) and the SI text, explaining the reasoning motivating each empirical exercise. We believe the draft is much clearer and thank the referee for helping improve these areas in particular.

Reviewer 3

We thank the referee for all the positive comments and support for the paper. They have substantially improved our paper. We address each point individually below (your comments in **bold**, ours in plain text, and excerpts from the paper indented).

I was happy to review this manuscript. It is nice to see another paper in the emerging climate-nutrition literature, and the focus on ENSO appears novel relative to other pertinent studies. The analysis is generally well-executed, with many of my questions about methodology already anticipated in the supplementary materials. The attention to spatial variation is particularly helpful. The paper is nicely written and the results are discussed clearly and with helpful visualizations. In the context of these strengths, I have a few comments and questions:

- 1. Some additional justification for why the focus on ENSO – rather than the more proximate determinants of temperature and precipitation – would be helpful. In other words, if ENSO influences nutrition via changes in temperature and precipitation, why not just model those relationships? I suspect the authors have a good answer, but a sentence or two in the text would help.**

We thank the referee for the opportunity to clarify our aim and scope. The paper now emphasizes the value of measuring the effect of ENSO itself. First, ENSO is arguably the most important planet-wide climate phenomenon at interannual scales, and is also predictable out to a horizon of a few months. Our paper aims to underscore that the wellbeing of a large part of humanity is tied to this predictable environmental process, and we are the first to estimate the effect on this aspect of human health at global scale. We argue that this is of inherent interest not only to the scientific community but also to institutions that plan around ENSO events (for example, the International Red Cross).

For these reasons, ENSO has been the object of several academic studies. While these mostly lack a global focus, they have proved that ENSO has destabilizing effects on agriculture^{28,29}, economic production³⁰, and social stability³¹ throughout teleconnected areas of the global tropics, and that ENSO affects human health outcomes directly through vector- and water-borne

²⁸ Iizumi, T., J.-J. Luo, A.J. Challinor, G. Sakurai, M. Yokozawa, H. Sakuma, M.E. Brown and T. Yamagata, "Impacts of El Niño Southern Oscillation on the global yields of major crops," *Nature Communications* 5, no. 3712, (2014).

²⁹ David, M., "Late Victorian holocausts: El Niño famines and the making of the third world," London: Verso, 2002.

³⁰ Hsiang, S. M. and Meng, K.C. "Tropical Economics." *American Economic Review: Papers & Proceedings* 2015, 105(5): 257-261.

³¹ Hsiang, S., K. Meng and M. Cane. "Civil conflicts are associated with the global climate," *Nature*, vol. 476, pp. 438-441. 2011.

infectious diseases^{32,33,34,35,36}, and indirectly through its effects on agricultural output³⁷. We contend that measuring the total effects of ENSO across all of these mechanisms at a global scale is an important contribution to this growing academic literature.

The following sentence in the introduction articulates the value of studying ENSO directly (line 30): “Given that probabilistic forecasts of ENSO have skill at predicting conditions months in advance, there is an opportunity to decouple food insecurity and human nutrition from this predictable climate process.” We have added a sentence to the discussion of the paper to restate the value of estimating ENSO’s effects, and thank the referee for encouraging us to do so. The sentence is (line 237): “Our work identifies a component of global undernutrition that society can predict, and can contribute to development of hunger early warning systems that would allow actors to deploy nutrition and humanitarian support operations in proactive instead of reactive ways.”

2. How is the analytic sample influenced by migration (e.g., recent arrivals in the DHS clusters, climate-related out-migration)? Some acknowledgement of the potential (unavoidable) biases would be helpful.

Thank you for raising this concern. We now discuss these as possible sources of bias at the end of the results section (line 185):

While our estimate of ENSO’s effect on child undernutrition is robust, there are nonetheless limitations imposed by both the nature of the data and structure of this research design. DHS data only selectively report migration, making it difficult to deal with any possible migration into or out of the sample that might occur in response to the ENSO cycle. Sufficiently severe ENSO events may also differentially influence likelihood of being in the sample, both at local scale, where e.g., worse-impacted children may be less likely to end up surveyed due to mortality or illness, and at larger scales, where events such as civil conflict that are known to respond to ENSO (Hsiang Meng and Cane, 2011) may plausibly inhibit the DHS’s ability to gather data or ensure data quality. While these aspects of sample selection may lead to unavoidable biases in our results, missing more vulnerable populations would likely bias us away from

³² M. Thomson, S. Mason, T. Phindela and S. Connor, "Use of rainfall and sea surface temperature monitoring for malaria early warning in Botswana," *Am. J. Trop. Med. Hyg.*, vol. 73, no. 1, pp. 214-221, 2005.

³³ World Health Organization, "El Niño and Health," WHO, Geneva, 1999.

³⁴ B. Cazelles, M. Chavez, A. McMichael and S. Hales, "Nonstationary influence of El Niño on the synchronous dengue epidemics in Thailand," *PLoS Med*, vol. 2, no. e106, 2005.

³⁵ M. Pascual, X. Rodo, S. Ellner, R. Colwell and M. Bouma, "Cholera dynamics and El Niño-Southern Oscillation," *Science*, vol. 289, pp. 1766-1769, 2000.

³⁶ A. Bennett, L. Epstein, R. Gilman, V. Cama, C. Bern, L. Cabrera, A. Lescano, J. Patz, C. Carcamo, C. Sterling and W. Checkley, "Effects of the 1997-1998 El Niño episode on community rates of diarrhea," *Am J Public Health*, vol. 102, pp. e63-69, 2012.

³⁷ C. Rosenzweig and D. Hillel, *Climate Variability and the Global Harvest: Impact of El Niño and Other Oscillations of Agro-ecosystems*, Oxford Univ. Press, 2008.

finding an effect of ENSO on health. Moreover, the consistency of the result across specifications and subsample suggests that their influence on the overall result is likely small.

3. Additional questions on the analytic sample: How many cases are dropped because of implausible or missing anthropometrics? Does this vary by country or region? Do the authors adjust for measurement error in birth timing (see Larsen et al. 2019 in Demography)?

We thank the referee for raising this concern. The total number of observations from the main sample that we flag and drop for implausible weight values is relatively small, with 0.56% of the sample being flagged for WAZ (see table below). Height error rates are higher, with HAZ having a 2.32% flag rate in the main sample, a pattern that is well-documented within the DHS and a general problem in child anthropometrics due to the difficulty and variability in measuring child length / height, especially for young children³⁸. This general relationship does not vary appreciably by UNICEF region, which sees average WAZ flag rates from 0.31% - 0.65%, and average HAZ between 0.94% - 2.93%, although there is some variation in both at country level.

Saliently, we test for and do not find that the rate of flagged observations varies with ENSO. The table “Anthropometric flags as a function of ENSO” below uses our main specification on contemporaneous effects on WAZ (Table 1, col 1) and our specification looking at lagged effects on HAZ among older children (Table S8, col 1), with both now expanded to include flagged observations and with the dependent variable being the probability that a child-level observation was flagged. Neither the contemporaneous or lagged specifications show an association of ENSO with changes in the probability that a child observation has flagged anthropometrics. We now speak directly to this issue in the Methods, where we have expanded the following sentence (line 307): “We calculated children's anthropometrics according to the WHO Anthro Child Growth Standard, excluding the small number of observations (0.56% for WAZ) that are flagged for having improbable values.”

³⁸ Pullum, Thomas W., Courtney Allen, Sorrel Namaste, and Trevor Croft. 2020. The Sensitivity of Anthropometric Estimates to Errors in the Measurement of Height, Weight, and Age for Children Under Five in Population-Based Surveys. DHS Methodological Reports No. 28. Rockville, Maryland, USA: ICF.
<https://dhsprogram.com/publications/publication-MR28-Methodological-Reports.cfm>

Table: WAZ and HAZ flag prevalence

	WAZ flags (%)	HAZ flags (%)
Full sample	0.56%	2.32%
UNICEF Regions:		
East Asia & Pacific	0.31%	2.93%
Latin America & Caribbean	0.43%	0.94%
Middle East & North Africa	0.60%	1.86%
South Asia	0.49%	2.36%
sub-Saharan Africa	0.65%	2.85%
East Asia & Pacific		
Cambodia	0.24%	1.73%
Timor Leste	0.39%	4.31%
Latin America & Caribbean		
Bolivia	0.60%	1.53%
Brazil	3.11%	7.67%
Colombia	0.16%	0.24%
Dominican Republic	0.75%	1.31%
Guatemala	1.22%	1.69%
Haiti	0.23%	0.83%
Honduras	0.17%	0.35%
Nicaragua	0.75%	2.04%
Peru	0.11%	0.34%
Middle East & North Africa		
Morocco	0.63%	2.09%
Yemen	0.45%	0.40%
sub-Saharan Africa		
Angola	0.34%	1.03%
Benin	0.66%	4.83%
Burkina Faso	0.56%	2.17%
Burundi	0.11%	0.69%
Cameroon	0.27%	1.51%
Central African Republic	0.98%	1.72%
Chad	0.44%	1.87%
Comoros	1.27%	4.63%
Congo, Rep.	0.10%	1.18%
Congo, Dem. Rep.	0.32%	3.90%
Cote d'Ivoire	0.54%	1.34%
Ethiopia	0.40%	2.29%
Gabon	0.18%	1.40%
Ghana	0.75%	1.91%
Guinea	1.30%	2.60%
Kenya	0.21%	1.38%
Lesotho	0.45%	2.54%
Liberia	1.11%	2.96%
Madagascar	0.39%	1.65%
Malawi	0.47%	3.13%
Mali	0.89%	2.86%
Mozambique	0.24%	1.89%
Namibia	0.63%	1.67%
Niger	0.52%	2.33%
Nigeria	1.77%	7.86%
Rwanda	0.31%	1.50%
São Tomé e Príncipe	0.42%	6.39%
Senegal	0.80%	1.18%
Sierra Leone	2.11%	7.36%
Swaziland	0.19%	1.90%
Tanzania	0.34%	1.49%
Togo	0.47%	1.21%
Uganda	0.56%	1.61%
Zambia	0.25%	1.84%
Zimbabwe	0.57%	2.30%
South Asia		
Bangladesh	0.50%	1.67%
India	0.50%	2.46%
Maldives	0.33%	2.12%

Table: Anthropometric flags as a function of ENSO

		(1) Weight for age flag	(2) Height for age flag
Year of survey	May-Dec Mean Niño 3.4 (°C)	-0.000290	
	P-value:	0.841	
	* I(> 50% Pos. Precip. Corr.)	0.000616	
	P-value:	0.531	
Year of birth	May-Dec Mean Niño 3.4 (°C)		0.00109
	P-value:		0.267
	* I(> 50% Pos. Precip. Corr.)		-0.000459
	P-value:		0.805
Year after birth	May-Dec Mean Niño 3.4 (°C)		0.00180
	P-value:		0.128
	* I(> 50% Pos. Precip. Corr.)		0.00162
	P-value:		0.439
Two years after birth	May-Dec Mean Niño 3.4 (°C)		0.000325
	P-value:		0.756
	* I(> 50% Pos. Precip. Corr.)		0.00242
	P-value:		0.332
Dependent variable mean:		0.00556	0.0162
Observations		1,260,186	675,631
R-squared		0.020	0.024

Lastly, we very much appreciate the note on timing per Larsen et al. (2019), though we believe this paper is largely unaffected for multiple reasons. First, their model focuses on how age misreporting influences HAZ as a function of earlier treatments, while ours is concerned with several different weight measures (WAZ, WHZ, BMI) as functions of contemporaneous treatment. Second, the Dec-Jan “calendar-year” bias they document would be a greater concern if we were assigning treatment differently for children early vs late in the calendar year, but we are assigning treatment by the May-April tropical year. Third, our result shows up in both younger and older children, despite the fact that the “round age anomaly” Larsen et al. document is heavily concentrated in infants. Lastly, our main models include month of interview by UNICEF region fixed effects as a baseline specification, which should control for a variety of possible seasonal patterns related to calendar month of survey. We nevertheless perform a simple test to see whether Larsen et al. -style concerns are influencing our results by taking our main WAZ and HAZ results and comparing them to the same specification with country-specific birth month fixed effects, which should absorb any of the country-specific intra-annual variation they document. The results, shown in Table “WAZ and HAZ with and without Fixed Effects by Month of Birth”, below, are nearly identical. We thus conclude there is little concern our results are biased by either of these two statistical artifacts.

Table: WAZ and HAZ with and without Fixed Effects by Month of Birth

		(1)	(2)	(3)	(4)
		WAZ	WAZ Country X Birth month FE	HAZ	HAZ Country X Birth month FE
Year of survey	May-Dec Mean Niño 3.4 (°C)	-0.0251**	-0.0245**		
	P-value:	0.0234	0.0260		
	* I(> 50% Pos. Precip. Corr.)	0.0733***	0.0715***		
	P-value:	0.00239	0.00222		
Year of birth	May-Dec Mean Niño 3.4 (°C)			-0.0327***	-0.0310***
	P-value:			0.00890	0.00548
	* I(> 50% Pos. Precip. Corr.)			0.0569**	0.0622**
	P-value:			0.0349	0.0302
Year after birth	May-Dec Mean Niño 3.4 (°C)			-0.0213*	-0.0195**
	P-value:			0.0668	0.0457
	* I(> 50% Pos. Precip. Corr.)			0.109	0.118
	P-value:			0.590	0.626
Two years after birth	May-Dec Mean Niño 3.4 (°C)			-0.0244**	-0.0232**
	P-value:			0.0266	0.0116
	* I(> 50% Pos. Precip. Corr.)			-0.00408	0.00439
	P-value:			0.906	0.890
Dependent variable mean:		-1.078	-1.078	-1.731	-1.731
Observations		1,253,176	1,253,176	664,658	664,658
R-squared		0.129	0.133	0.136	0.144

4. The authors report that height-for-age did not respond to ENSO (p. 4). However, this is to be expected if they applied the same empirical strategy as for WHZ. I'd recommend they model HAZ among older children (e.g., 3-5 years) as a function of early life ENSO (see Alderman and Headey 2018 in PLOS for discussion of timing; and a number of recent papers on climate and nutrition have adjusted their samples and exposure measures accordingly).

We are very grateful for this suggestion, echoed by referee 1. We follow the referees' advice and now include Table S8 which estimates the anthropometric effects of early life exposure to ENSO in 24-59 month old children. Specifically, we assign birth dates to tropical year using the same May-April spring barrier assignment used for interview dates, and estimate a version of our main precipitation teleconnection-differentiated model where HAZ and stunting are a function of ENSO state during the year of birth, as well as each of the subsequent two years of the child's life.

We find that early life exposure leads to persistent effects, with children experiencing warmer, more El-Niño-like conditions during their year of birth and two following years having lower HAZ and being more likely to be undernourished when surveyed at ages 2-5. This effect is mirrored and opposite in the small number of positively teleconnected regions for the first year of life, but then seems to shift, with less evidence of stunting. We believe these results substantially strengthen the evidence of ENSO's malnutrition effects and highlight their persistent nature, which we now highlight in the main paper (line 162: "On the other hand, child stunting remains affected years after negative shocks from ENSO (Table S8), consistent with

height being slower to respond to health shocks than weight and with the first two years of life being the riskiest period for growth faltering due to scarring”). We are very grateful to the referees for the suggestion to dig deeper.

		(1) Height for age	(2) Probability below WHO Standard for stunted
Year of birth	May-Dec Mean Niño 3.4 (°C)	-0.0327***	0.00903**
	p-value:	0.00890	0.0351
	* I(> 50% Pos. Precip. Corr.)	0.0569**	-0.00157
	p-value:	0.0349	0.821
Year after birth	May-Dec Mean Niño 3.4 (°C)	-0.0213*	0.00724*
	p-value:	0.0668	0.0838
	* I(> 50% Pos. Precip. Corr.)	-0.0248	0.00913
	p-value:	0.590	0.386
Two years after birth	May-Dec Mean Niño 3.4 (°C)	-0.0244**	0.00774*
	p-value:	0.0266	0.0610
	* I(> 50% Pos. Precip. Corr.)	-0.00408	0.00325
	p-value:	0.906	0.711
Dependent variable mean:		-1.713	0.417
Observations		664,658	664,658
R-squared		0.136	0.120

5. The discussion section refers to statistics on “undernourishment”, which are quite different from wasting or stunting. In my view, it would be more helpful to stick strictly to a discussion of global wasting/WHZ (the main outcome in the analysis) or be very clear about the differences between what is measured in the analysis and “undernourishment”.

We agree with this point and have changed all mentions of “undernourishment” to “underweight” following WHO nomenclature for < -2 SD in weight for age.

6. Invoking “famine” in the discussion is problematic in my view. All of the formally-declared famines in the past few decades have occurred in conflict situations, making attribution to ENSO problematic (in my view). Even in earlier centuries these relationships are problematic. Indeed, Mike Davis – who the authors cite – made the explicit point in *Late Victorian Holocausts* that ENSO only causes mass famine when the social and political conditions allow it.

We appreciate the nuance of this view, and accept that while ENSO may be driving food insecurity, a Sennian³⁹ view of famines would insist that they must involve at least some degree of organizational failure leading to deprivation of food for vulnerable groups. We have thus changed each mention of famine as follows: “episodic food insecurity in the tropics” (line 232) rather than “episodic famine”, “shocks like COVID-19 capable of generating extensive food insecurity” (line 236) rather than “famine”, and (line 231) “Indeed, the 2015 El Niño likely played a major role in worsening global hunger” rather than “famine”.

- 7. I’m surprised by the lack of engagement with the existing literature on climate and nutrition. As just one example - while I appreciate Solomon Hsiang’s work - it is inexplicable that he is the most-cited author in this paper when his work has not focused on nutrition to my knowledge. Upshot: please engage with this literature, especially recent papers that have also pooled large numbers of DHS samples to look at similar issues (e.g., Cooper et al. 2019 in PNAS, Grace et al. 2015 in GEC, Thiede et al. 2020 in GEC).**

We thank the referee for this comment, and welcome the opportunity to better link our work to related nutrition literature that uses the DHS. This comment provided us with an impetus to add a number of citations beyond the interdisciplinary climate impacts literature. We have modified the following sentence in the introduction to read as follows (line 38):

“We leverage over one million child anthropometric records spanning four decades and all developing country regions, building on a growing literature using the Demographic and Health Surveys to document the effect of weather variation on child nutrition (Baker and Anttila-Hughes, 2020; Thiede and Strube, 2020; Cooper et al., 2019, Grace et al., 2015).”

The number of citations in this revision has increased by more than 30%. In addition to those mentioned above, we have expanded on the citations relating to the environment-nutrition link (Del Ninno and Lundberg, 2005; Alderman and Heady, 2018), and added a number that relate to the potential agricultural mechanism behind our results (Schlenker and Roberts, 2009; Schlenker and Lobell, 2010).

³⁹ Sen, A. (1981). “Ingredients of famine analysis: availability and entitlements.” *The Quarterly Journal of Economics*, 96(3), 433-464.

REVIEWER COMMENTS

Reviewer #1 (Remarks to the Author):

The manuscript reinforces the evidence of ENSO's consequences on child malnutrition. The additional analyses incorporated, and the new findings related to height-for-age and stunting solidified the conclusions of the previous manuscript version.

Definitely, the article contributes to the study of children's health outcomes in the context of the ENSO phenomenon and suggests the urgency of social & health policies to cushion ENSO's consequences.

Thank you for addressing all the comments. I am pleased to see that some suggestions were helpful. Finally, I recommend the publication of this article.

Reviewer #2 (Remarks to the Author):

Thank you for your thorough and thoughtful work in response to my comments. You have done a very nice job addressing most of my feedback and concerns about the original manuscript. However, there are still a few issues that I believe need to be addressed.

Most importantly, while I appreciate your extended discussion related to mechanisms, I feel that the language used and the conclusions drawn are not appropriate given your methods, data, and findings. I have discussed specific instances of this below. I recommend that the authors revisit the language throughout the paper to make sure they are not implying causal relationships, as that would not be appropriate for this study design (again, see examples below).

Second, I appreciate your motivation for comparing your results to other public health interventions. However, I suggest making it even clearer in the text that you are using a "back of the envelope" calculation that is a rough estimate. My concern is that the number may be taken out of context as one of your findings, when it is not, in fact, the main focus of your paper.

Specific Comments:

In your new discussion paragraph:

"The consistency across regions suggests that local context is not generating first-order differences in how ENSO affects children's health..."

This statement is likely not true and misleading. As you stated, you are looking at the net effect of ENSO on many different mechanistic pathways. Hence, the same outcome (i.e. a change in malnutrition associated with ENSO) could be observed through many different causal pathways. It is misleading, and problematic, to claim that a consistent relationship across locations implies that local context is not important. This is simply not true and is not supported by your data.

"The importance of precipitation correlation in determining ENSO's effect suggests that agriculture plays a strong role in linking ENSO to nutrition outcomes. Other potential channels, such as vector-borne or water-borne diseases, would likely increase with higher precipitation, as would impacts of flooding."

Similar to my comment above, this statement is misleading and unsupported by your findings. The relationships between meteorology/hydrology and vector-borne and water-borne diseases are

complex and dependent on local variables such as water and sanitation systems, healthcare systems, cultural practices, etc. In fact, there are many instances where decreases in rainfall (e.g., droughts) lead to increased diarrheal disease (a primary driver of childhood malnutrition) [see <https://doi.org/10.1289/EHP6181> for more information]. I do not question that precipitation is an important mediator of the ENSO-malnutrition relationship. And I agree that agriculture is likely an important mediator, but your findings do not exclude other mediators such as vector-borne and water-borne diseases.

"Finally, the fact that the effect of a warmer ENSO state varies by the direction of precipitation correlation suggests that the health effects are primarily operating through the precipitation channel and not the increase in temperature, since all places experience a temperature increase."

This statement is also misleading. While many locations do experience an increase in temperature associated with El Niño events, the magnitude of those temperature changes is variable and the health impacts of temperature changes are dependent on local factors. The observation that precipitation is an important mediating variable does not imply that temperature is not an important variable, especially as you have not explicitly studied temperature changes in this analysis.

Other examples of language that I feel should be modified:

The use of the word "drives" in the title is a bit too strong as it implies causality. I would suggest something such as "ENSO [impacts/is associated with/and] childhood undernutrition..."

Line 20: "the 2015 El Niño pushed almost 6 million children into underweight status..." The term "pushed" here implies causality.

Reviewer #3 (Remarks to the Author):

Thank you for the opportunity to review the revised version of the manuscript. I reviewed the initial submission and in my view the authors did an excellent job addressing my comments and (again, in my view) the comments of the other reviewers. Based on my reading, this paper has the potential to significantly contribute to the literature on climate/environmental variability and malnutrition. The focus on ENSO is novel and has considerable value-added vis a vis prior studies' focus on temperature and precipitation. Thanks to the authors for this excellent work, including extensive revisions. I have just two minor notes below:

- (1) In addition to reporting the number of cases with implausible anthropometric values, it would be helpful to also report the number *missing* these values
- (2) ~line 214: a citation or two regarding climate and water/vector-borne diseases could be helpful

Reviewer 2

Thank you for your thorough and thoughtful work in response to my comments. You have done a very nice job addressing most of my feedback and concerns about the original manuscript. However, there are still a few issues that I believe need to be addressed.

Most importantly, while I appreciate your extended discussion related to mechanisms, I feel that the language used and the conclusions drawn are not appropriate given your methods, data, and findings. I have discussed specific instances of this below. I recommend that the authors revisit the language throughout the paper to make sure they are not implying causal relationships, as that would not be appropriate for this study design (again, see examples below).

We thank the referee for the positive comments and support for the paper. All of your comments below are well-taken. Below, we address each point individually (your comments in **bold**, ours in plain text, and excerpts from the paper indented).

Before discussing individual points, we also wish to address overarching points that are present in a number of the individual comments. Most importantly, we agree with the referee's advice on our discussion of mechanisms and have rephrased that entire paragraph to avoid unsupported precision. We have also changed several instances of language in the paper following your advice, most prominently on the title of the paper.

Further, the response leads us to believe that we have been too indirect about the fact that we do not think ENSO is the main driver, and far from the sole determinant, of the health status *levels* in our area of study. Rather, what we identify is the *change* in local average health status due to ENSO events. Second, and relatedly, we of course firmly agree that local context matters for determinants of overall average health of a population. To that end, and in response to your comments, we have added the following sentences to the introduction on lines 44-48, immediately after we describe our research and before we discuss the results:

Our research design estimates the change in nutritional status associated with being in a positive or negative ENSO state compared to a counterfactual of ENSO-neutral conditions. Our results describe shocks to nutritional status rather than identifying the average level of health in a location, which is a complex function of local conditions, such as infrastructure, policies, and the environment.

We have also changed the last sentence in the abstract (the subject of a comment discussed below) from “demonstrating the degree to which human well-being remains subject to

predictable climatic processes” to “demonstrates **a pathway through which** human well-being remains subject to predictable climatic processes” [emphasis only in this response].

We hope that these changes, in addition to the various others discussed below, reinforce what we take to be the contribution of this paper. Namely, identifying one component of child nutrition and health that systematically varies with a predictable global, natural cycle in the climate and may provide both new information on a determinant of ill-health and guidance towards a policy response.

Second, I appreciate your motivation for comparing your results to other public health interventions. However, I suggest making it even clearer in the text that you are using a "back of the envelope" calculation that is a rough estimate. My concern is that the number may be taken out of context as one of your findings, when it is not, in fact, the main focus of your paper.

We thank the referee for this suggestion, and have updated the language accordingly: “To give context to the size of these effects, we provide illustrative order-of-magnitude calculations of...” (line 123) and “...would require approximately 134 million...” (line 133)

Specific Comments:

In your new discussion paragraph: "The consistency across regions suggests that local context is not generating first-order differences in how ENSO affects children's health..." This statement is likely not true and misleading. As you stated, you are looking at the net effect of ENSO on many different mechanistic pathways. Hence, the same outcome (i.e. a change in malnutrition associated with ENSO) could be observed through many different causal pathways. It is misleading, and problematic, to claim that a consistent relationship across locations implies that local context is not important. This is simply not true and is not supported by your data.

We thank the referee for pointing this out. We appreciate the nuance of the referee's point, and would like emphasize that we are not arguing that local context is not relevant for health. In order to avoid misleading claims we have removed the relevant sentence from the discussion.

"The importance of precipitation correlation in determining ENSO's effect suggests that agriculture plays a strong role in linking ENSO to nutrition outcomes. Other potential channels, such as vector-borne or water-borne diseases, would likely increase with higher precipitation, as would impacts of flooding." Similar to my comment above, this statement is misleading and unsupported by your findings. The relationships between meteorology/hydrology and vector-borne and water-borne diseases are complex and

dependent on local variables such as water and sanitation systems, healthcare systems, cultural practices, etc. In fact, there are many instances where decreases in rainfall (e.g., droughts) lead to increased diarrheal disease (a primary driver of childhood malnutrition) [see <https://doi.org/10.1289/EHP6181> for more information]. I do not question that precipitation is an important mediator of the ENSO-malnutrition relationship. And I agree that agriculture is likely an important mediator, but your findings do not exclude other mediators such as vector-borne and water-borne diseases.

We agree with the referee's points, and have rewritten the relevant paragraph to reflect the referee's suggestions (lines 205-219):

This analysis measures the total effect of ENSO on child nutrition through all potential measurements and for all affected countries with available data. The negative relationship between child nutrition and warm ENSO state does not appear to vary appreciably across space, with the effects for major world regions in the sample being statistically indistinguishable from our main effect (Fig. 2C). The fact that the effect of a warmer ENSO state varies by the direction of precipitation correlation highlights the importance of precipitation as a mediator in the ENSO-malnutrition relationship. The importance of precipitation may indicate that agriculture plays a strong role in linking ENSO to nutrition outcomes, though we cannot reject the possibility that other channels play important roles in some locations. Other potential channels, such as vector-borne diseases, would likely harm health when precipitation is higher,³⁹ as would impacts of flooding. On the other hand, decreases in rainfall can lead to increased diarrheal disease.⁴⁰ Since other channels are not translating into adverse health outcomes for children in places where precipitation increases during the El Niño state, the improved nutrition observed in the data in these locations suggests that increases in agricultural production due to higher rainfall may be a key channel in the global ENSO-nutrition relationship.

"Finally, the fact that the effect of a warmer ENSO state varies by the direction of precipitation correlation suggests that the health effects are primarily operating through the precipitation channel and not the increase in temperature, since all places experience a temperature increase." This statement is also misleading. While many locations do experience an increase in temperature associated with El Niño events, the magnitude of those temperature changes is variable and the health impacts of temperature changes are dependent on local factors. The observation that precipitation is an important mediating variable does not imply that temperature is not an important variable, especially as you have not explicitly studied temperature changes in this analysis.

We agree with the referee that the importance of precipitation as a mediating variable does not obviate the importance of temperature. We have removed the sentence in question.

Other examples of language that I feel should be modified:

The use of the word "drives" in the title is a bit too strong as it implies causality. I would suggest something such as "ENSO [impacts/is associated with/and] childhood undernutrition..."

We thank the referee for the suggestion. We have changed the title to “ENSO Impacts Child Undernutrition in the Global Tropics,” and agree that this better captures the conclusions of the paper.

**Line 20: "the 2015 El Niño pushed almost 6 million children into underweight status..."
The term "pushed" here implies causality.**

We have removed the word “pushed” and rephrased the sentence as follows: “Results imply that almost 6 million additional children were underweight during the 2015 El Niño compared to a counterfactual of neutral ENSO conditions in 2015.” This phrasing is a more literal reflection of the econometric approach taken in the paper.

Reviewer 3

Thank you for the opportunity to review the revised version of the manuscript. I reviewed the initial submission and in my view the authors did an excellent job addressing my comments and (again, in my view) the comments of the other reviewers. Based on my reading, this paper has the potential to significantly contribute to the literature on climate/environmental variability and malnutrition. The focus on ENSO is novel and has considerable value-added vis a vis prior studies' focus on temperature and precipitation. Thanks to the authors for this excellent work, including extensive revisions.

We thank the referee for the positive comments and support of the paper.

I have just two minor notes below:

(1) In addition to reporting the number of cases with implausible anthropometric values, it would be helpful to also report the number *missing* these values

We thank the referee for the suggestion, and now report that number: "...and those with missing values (0.57% for WAZ)." (line 310-311).

(2) ~line 214: a citation or two regarding climate and water/vector-borne diseases could be helpful

Following the referee's good suggestion, we now cite literature regarding the relationship between precipitation, vector-borne diseases, and diarrheal disease: "Other potential channels, such as vector-borne diseases, would likely harm health when precipitation is higher (McCord and Anttila-Hughes, 2017), as would impacts of flooding. On the other hand, decreases in rainfall can lead to increased diarrheal disease (Kraay et al., 2020)." (line 213-215)

References

- McCord, Gordon C., and Jesse K. Anttila-Hughes. "A malaria ecology index predicted spatial and temporal variation of malaria burden and efficacy of antimalarial interventions based on African serological data." *The American journal of tropical medicine and hygiene* 96.3 (2017): 616-623.
- Kraay, Alicia NM, et al. "Understanding the Impact of Rainfall on Diarrhea: Testing the Concentration-Dilution Hypothesis Using a Systematic Review and Meta-Analysis." *Environmental health perspectives* 128.12 (2020): 126001.

REVIEWERS' COMMENTS

Reviewer #2 (Remarks to the Author):

Thank you for your responses to my comments. I believe the language is much more appropriate after your revision. Congratulations on a very good paper. I only have one remaining minor comment:

I do not understand the meaning of the sentence below. Could you rewrite this sentence to improve clarity?

"Since other channels are not translating into adverse health outcomes for children in places where precipitation increases during the El Niño state, the improved nutrition observed in the data in these locations suggests that increases in agricultural production due to higher rainfall may be a key channel in the global ENSO-nutrition relationship."

Response to reviewer comments

Reviewer #2 (Remarks to the Author):

Thank you for your responses to my comments. I believe the language is much more appropriate after your revision. Congratulations on a very good paper. I only have one remaining minor comment:

Thank you very much for all of your work helping to make this a better, stronger paper.

I do not understand the meaning of the sentence below. Could you rewrite this sentence to improve clarity?

"Since other channels are not translating into adverse health outcomes for children in places where precipitation increases during the El Niño state, the improved nutrition observed in the data in these locations suggests that increases in agricultural production due to higher rainfall may be a key channel in the global ENSO-nutrition relationship."

We agree that the sentence is unwieldy and have replaced it with the following:

"Whatever channels may exist, the improved nutritional outcomes during the El Niño state in areas where precipitation increases suggests that negative effects from channels such as flooding and disease are on average outweighed by the positive effects from channels such as agriculture."